



# Distinct secondary ice production processes observed in radar Doppler spectra: insights from a case study

Anne-Claire Billault-Roux[1], Paraskevi Georgakaki[2], Josué Gehring[3], Louis Jaffeux[4], Alfons Schwarzenboeck[4], Pierre Coutris[4], Athanasios Nenes[2,5], and Alexis Berne[1]

[1]Environmental Remote Sensing Laboratory, École Polytechnique Fédérale de Lausanne, Lausanne, Switzerland
[2]Laboratory of Atmospheric Processes and their Impacts, École Polytechnique Fédérale de Lausanne, Lausanne, Switzerland
[3]Swiss Federal Office of Meteorology and Climatology, Geneva, Switzerland
[4]Laboratoire de Météorologie Physique, Aubière, France
[5]Center for the Study of Air Quality and Climate Change, Institute of Chemical Engineering Sciences, Foundation for Research and Technology Hellas, Patras, Greece

**Correspondence:** alexis.berne@epfl.ch, anne-claire.billault-roux@epfl.ch

**Abstract.** Secondary ice production (SIP) has an essential role in cloud and precipitation microphysics. In recent years, substantial insights were gained into SIP by combining experimental, modeling, and observational approaches. Remote sensing instruments, and among them meteorological radars, offer the possibility to study clouds and precipitation in extended areas over long time periods, and are highly valuable to understand the spatio-temporal structure of microphysical processes. Multi-modal Doppler spectra measured by vertically-pointing radars reveal the coexistence, within a radar resolution volume, of hydrometeor populations with distinct properties; as such, they can provide decisive insight into precipitation microphysics. This paper leverages polarimetric radar Doppler spectra as a tool to study the microphysical processes that took place during a snowfall event on 27 January 2021, in the Swiss Jura Mountains, during the ICE GENESIS campaign. A multi-layered cloud system was present, with ice particles sedimenting through a supercooled liquid water (SLW) layer in a seeder-feeder configuration. Building on a Doppler peak detection algorithm, we implement a peak labeling procedure to identify the particle type(s) that may be present within a radar resolution volume. With this approach, we can visualize spatio-temporal features in the radar time series that point to the occurrence of distinct mechanisms at different stages of the event. By focusing on three 30-minute phases of the case study, and by using the detailed information contained in the Doppler spectra, together with dual-frequency radar measurements, aircraft in-situ images, and simulated profiles of atmospheric variables, we narrow down the possible processes which can be responsible for the observed signatures. Depending on the availability of SLW and the droplet sizes, on the temperature range, and on the interaction between the liquid and ice particles, various SIP processes are identified as plausible, with distinct fingerprints in the radar Doppler spectra. A simple modeling approach suggests that the ice crystal number concentrations likely exceed typical concentrations of ice nucleating particles by one to four orders of magnitude. While a robust proof of occurrence of a given SIP mechanism cannot be easily established, the multi-sensor data provides various independent elements each supporting the proposed interpretations.



# 1 Introduction

Mixed-phase clouds (MPCs), in which ice crystals and snow particles coexist with supercooled liquid water (SLW) droplets, have a key role in the atmosphere both in terms of their impact on the Earth's radiation budget (McCoy et al., 2016; Matus and L'Ecuyer, 2017) and on precipitation processes (Mülmenstädt et al., 2015). MPCs are intrinsically unstable structures: without a sustained source of supercooled liquid water, the liquid phase tends to be depleted through the Wegener-Bergeron-Findeisen process or riming (Korolev et al., 2017), leading to a full glaciation of the cloud. They are, however, very frequently observed (e.g., in the Arctic, Intrieri et al., 2002 or in orographic terrain, Lohmann et al., 2016) and there exist several means by which the liquid water content is sustained: frontal or orographic lifting of the air masses, for instance, are associated with vertical velocities sufficient to maintain supersaturation with respect to liquid water (Korolev and Field, 2008; Georgakaki et al., 2021); small-scale vertical motion caused by turbulence (Field et al., 2014), as well as cloud top radiative cooling (Morrison et al., 2012), also enable the formation of supercooled cloud droplets.

Among the processes that occur in the mixed phase, the production of ice through secondary processes has received substantial attention in recent years. Secondary ice production (SIP) is defined by contrast to primary ice production, through which, at temperatures warmer than -38°C, ice crystals are formed via heterogeneous nucleation requiring active ice nucleating particles (INP). SIP is thought to increase the ice crystal number concentration (ICNC) by up to several orders of magnitude, which impacts the phase partitioning in MPCs (Atlas et al., 2020) and the resulting overall radiation budget (Sun and Shine, 1994; Young et al., 2019) and precipitation (Luke et al., 2021; Dedekind et al., 2022). Several processes have been identified through which ice multiplication can occur (Field et al., 2017; Korolev and Leisner, 2020), among which three are typically cited as dominant. The so-called *Hallett-Mossop (HM) rime splintering mechanism* (Hallett and Mossop, 1974) occurs as supercooled cloud droplets or drizzle/rain drops rime onto ice particles, generating ice splinters in the process; HM is active between -8°C and -3°C, with a maximum efficiency around -5°C. Secondary ice particles can also be produced when supercooled drops shatter into several fragments when freezing upon contact with an ice particle or an INP (e.g., Takahashi and Yamashita, 1977; Phillips et al., 2018). This *droplet shattering* process requires drizzle-size drops of at least 50 $\mu$m (Wildeman et al., 2017); certain studies have suggested that the process is more efficient for larger drops ($\gtrsim$ 300 $\mu$m, Lauber et al., 2018; Keinert et al., 2020; Kleinheins et al., 2021), which could produce a larger number of fragments. Contrary to HM, it does not seem restricted to a clearly-established temperature range as it was reported to occur at both cold ($\sim$-15°C, Korolev and Leisner, 2020) and warmer temperatures (with the recirculation of rain drops above the melting layer, Korolev et al., 2020; Lauber et al., 2021). Ice-ice collisions, facilitated in turbulent regions or when ice particles have different settling velocities, can also produce secondary ice fragments (Vardiman, 1978; Takahashi et al., 1995; Schwarzenboeck et al., 2009). *Collisional break-up* is thought to be a substantial source of secondary ice particles in certain environments, such as wintertime alpine clouds (Dedekind et al., 2021), particularly under the frequent seeder-feeder cloud configurations observed in Switzerland (Grazioli et al., 2015; Proske et al., 2021; Georgakaki et al., 2022), although its underlying physical mechanisms are still debated (Korolev and Leisner, 2020). The presence of rimed particles is considered an important ingredient (Phillips et al., 2017a, b), based on the intuition



that these particles, with their higher mass and fall speed, are more likely to cause efficient breakup during high-kinetic-energy collisions with other ice particles.

  Proof of SIP mostly stems from in-situ observations showing that measured ICNCs considerably exceed values that would result from primary ice nucleation, controlled by the number concentration of active INPs (Mossop et al., 1970; Hobbs and

Rangno, 1985; Lloyd et al., 2015; Pasquier et al., 2022). Additional evidence was obtained in refined setups, which could verify that some snow crystals did not contain an INP (Mignani et al., 2019), and must have been generated through SIP. Such measurements remain sparse and these approaches are difficult to implement for a statistical characterization of SIP processes and their spatial and temporal dynamics. High-resolution modeling has helped improve their understanding (e.g., Sotiropoulou et al., 2020; Sullivan et al., 2018; Waman et al., 2022), but possible discrepancies with observations are difficult to interpret

due to the numerous hypotheses involved in the microphysical parameterizations (Sotiropoulou et al., 2021). Remote sensing observations, although indirect, provide valuable insight into cloud and precipitation processes in the entire atmospheric column. Passive sensors such as microwave radiometers allow estimating integrated quantities like the liquid water path (LWP, e.g., Löhnert and Crewell, 2003; Billault-Roux and Berne, 2021), which is relevant to monitor the formation and evolution of MPCs containing supercooled liquid cloud or drizzle droplets (e.g., Ramelli et al., 2021). Active remote sensing, mostly

with meteorological radars, is an additional popular tool for cloud and precipitation studies. Time series of radar moments can convey information on snowfall growth and decay (through the radar equivalent reflectivity factor, shortened as reflectivity, $Z_e$) or the occurrence of riming, visible through enhanced mean Doppler velocity (MDV).

Radar Doppler spectra from vertically-pointing profilers, which feature the reflectivity-weighted distribution of Doppler velocity in a radar volume, allow separating the contribution of fast- and slow-falling particles in the radar echo (e.g., Kollias

et al., 2002; Luke and Kollias, 2013; Kneifel et al., 2016). One particularly striking feature is when Doppler spectra, deviating from a Gaussian shape, have several distinct modes. This is usually the sign that hydrometeor populations with different microphysical properties are present in the same radar volume, as studied, for instance, by Zawadzki et al. (2001); Shupe et al. (2004); Kalesse et al. (2016). Depending on the properties of each peak (reflectivity and Doppler velocity), they may indicate that SLW droplets are present (Kalesse et al., 2016), or that new ice is formed. Additional information can be lever-

aged, when available, from spectral polarimetric measurements through the spectral linear depolarization ratio (LDR, e.g., Oue et al., 2018; Luke et al., 2021). While radar measurements alone are not sufficient to actually demonstrate the occurrence of SIP, some signatures can be identified that reasonably suggest such processes (Lauber et al., 2021; Luke et al., 2021; Li et al., 2021).

  In this study, we focus on a snowfall event that took place during the ICE GENESIS campaign (Billault-Roux et al., 2023b) in

the Swiss Jura, on 27 January 2021, during the passage of a warm front. A seeder-feeder configuration was observed, whereby ice particles sedimented through a SLW-containing cloud layer. Doppler spectra with persistent multi-modalities extending over several kilometers were recorded, pointing to the occurrence of complex microphysical processes. This is further supported by in-situ aircraft observations of ice and snow particles. An in-depth analysis of the signatures in the multi-sensor data and of atmospheric profiles obtained with high-resolution numerical modeling suggests that SIP was possibly taking place through



different mechanisms. The data and instrumentation are presented in Sect. 2, and the methods used for the analysis of the multi-modal spectra are detailed in Sect. 3. An overview of the event is provided in Sect. 4 with the synoptic context and an outline of the main observations. We then focus (Sect. 5) on three time frames where different signatures are observed, for which we propose interpretations.

## 2 Data and instrumentation

In this work, multi-sensor measurements are used to investigate microphysical processes during a snowfall event of the ICE GENESIS campaign (Billault-Roux et al., 2023b), which was conducted within the Swiss Jura Mountains, in January 2021. The deployment took place in La Chaux-de-Fonds (LCDF), at an altitude of 1020 m above mean sea level, which will be hereafter used as a reference: unless specified otherwise, altitudes will be expressed as a range, i.e., in m or km above ground level. The setup featured ground-based sensors, including an automatic weather station from the Swiss Federal Office of Meteorology and Climatology (MeteoSwiss) that provided measurements of standard meteorological variables and precipitation rate, and remote sensing instruments as detailed below. The ground instrumentation was complemented with in-situ probes on board a scientific aircraft that flew at various altitude levels above the ground site.

### 2.1 Ground-based remote sensing

We hereafter focus on data from two radars, whose settings are summarized in Table 1. WProf is a high-sensitivity, dual-polarization, frequency-modulated continuous wave (FMCW) W-band Doppler spectral zenith profiler (Küchler et al., 2017), operated in simultaneous transmit/receive mode. First moments (radar equivalent reflectivity factor $Z_{e,W}$ and mean Doppler velocity MDV) are used as well as full Doppler reflectivity spectra. The spectral slanted linear depolarization ratio (SLDR) is computed from the Doppler spectra measured in horizontal and vertical polarization and the covariance spectrum (Matrosov et al., 2012; Galletti et al., 2014; Myagkov et al., 2016; Ryzhkov and Zrnic, 2019). Note that the spectral SLDR measurements are only valid if the cross-polarized component of the received signal exceeds the noise level in the corresponding channel (Matrosov and Kropfli, 1993; Radenz et al., 2019). Attenuation due to water or snow accumulating on the radome is not considered an issue, as blowers were active during the entire measurement period, keeping the surface of the radome dry and snow-free. The W-band data used in this study are otherwise not corrected for path attenuation (see further on, Sect. 5). In addition to the radar variables, WProf allows retrieving estimates of LWP through the brightness temperature measured by a joint 89-GHz radiometer (Küchler et al., 2017; Billault-Roux and Berne, 2021). The error on retrieved LWP ($\sim$ 18%) increases during snowfall due to the radiative contribution of snow particles to the measured brightness temperature, but general trends in LWP are nevertheless considered reliable (Billault-Roux and Berne, 2021).

ROXI (Viltard et al., 2019) is an X-band single-polarization Doppler spectral zenith profiler. A cross-calibration of the radars was performed using independent measurements from a scanning X-band radar which had absolute calibration during the campaign (Billault-Roux et al., 2023a, and Appendix thereof). X-band reflectivity ($Z_{e,X}$) values are interpolated to the time and



| | WProf | ROXI |
|---|---|---|
| Frequency (GHz) | 94 | 9.4 |
| Transmission | FMCW, simultaneous transmit/receive | pulsed |
| 3-dB beam width ($°$) | 0.53 | 1.8 |
| Sensitivity (dBZ) [at range (km)] | -45 [0.5] / -41 [2] / -39 [5] | -19 [2] |
| Time resolution (s) | 5 | 3 |
| Range resolution (m) | 7.5 / 16 / 32 | 50 |
| Nyquist velocity (m s$^{-1}$) | 10.8 / 6.92 / 3.3 | 11 |
| Velocity resolution (m s$^{-1}$) | 0.02 / 0.014 / 0.013 | 0.1 |

**Table 1.** Properties and parameters of the ground-based and airborne radars. WProf uses three chirps, whose ranges are as follows: chirp 0: 104-998 m, chirp 1: 1008-3496 m, chirp 2: 3512-8683 m; when applicable, the properties for each chirp are separated by "/". The maximum range of ROXI is 6.4 km.

range resolution of WProf and used for computation of the dual-frequency ratio of reflectivity (DFR $= Z_{e,X}$-$Z_{e,W}$, with $Z_{e,W}$ and $Z_{e,X}$ in dBZ).

## 2.2 In-situ aircraft measurements

In-situ measurements of snowfall were conducted at various altitude levels by the scientific aircraft Safire-ATR42, equipped with a extensive set of probes as listed in Billault-Roux et al. (2023b) and, in particular, three different optical array probes which are used in this work. The high-volume precipitation spectrometer (HVPS) (resp. precipitation imaging probe, PIP, and 2D-Stereo, 2D-S) could collect images of particles with diameters ranging from 150 $\mu$m to 1.92 cm (resp. 100 $\mu$m to 6.4 mm, 10 $\mu$m to 1.28 mm). An automatic classification algorithm (Jaffeux et al., 2022) allows identifying particle habits from 2D-S and PIP images. In addition, cloud liquid water content (LWC) is estimated with a cloud droplet probe (CDP-2, Faber et al., 2018), which samples droplets up to 50 $\mu$m.

## 2.3 WRF model runs

Simulations of the case study were run with the Weather Research and Forecasting (WRF) model, version 4.0.1. Three two-way nested domains were used in a downscaling approach, with a horizontal resolution of respectively 12, 3, and 1 km (Appendix C1). The initial conditions and lateral forcing were obtained from the 6-hourly National Centers for Environmental Prediction (NCEP) Global Final Analysis (FNL) dataset at $1° \times 1°$ grid resolution. Other static fields were obtained from default WRF pre-processing system datasets at a resolution of 30-arc-seconds for both the topography and the land use fields. A grid spacing of 97 vertical eta levels was used, with a refined resolution of $\sim 100$ m up to mid-troposphere, following Vignon et al. (2021). The double-moment microphysical scheme of Morrison et al. (2005) (M05) was employed, following the implementation of





Georgakaki et al. (2022) (control run in the latter study). As the cloud droplets are represented with a single-moment approach in the M05 scheme, a constant droplet number concentration has to be considered. Here we set it to 50 cm$^{-3}$, consistent with CDP-2 measurements during the case study of interest. Additional physics options include the implementation of the Quasi-Normal Scale Elimination (QNSE) planetary boundary layer scheme (Sukoriansky et al., 2005), the Noah land surface scheme

and the Rapid Radiative Transfer Model for General Circulation Models (RRTMG) radiation scheme to model the shortwave and longwave radiative transfer. The Kain-Fritsch cumulus parameterization is also activated in the 12-km resolution domain. Atmospheric variables in the innermost domain were output with a 5-min time resolution, starting on 25 January at 12 UTC, allowing for a sufficient spin-up time before the onset of precipitation at the ground site in the early morning hours of 27 January. It was verified that the simulated WRF surface meteorological variables agreed reasonably well with weather sta-

tion measurements (Appendix Fig. C2). The WRF simulations are used in this study to provide high-resolution temperature, wind and humidity profiles, to gain an understanding of the mesoscale processes and how they may contribute to snowfall microphysics over LCDF.

## 3  Methods

### 3.1  Doppler spectra peak finding algorithm

In order to perform a systematic identification of multi-modalities in Doppler spectra, an automatic peak identification routine was implemented. The *pyPEAKO* code (Kalesse et al., 2019) was used after adjustment to our dataset. The algorithm is trained on a manually-labeled dataset, consisting of 300 WProf spectra for each chirp (no improvement was noted when including more spectra), randomly sampled from 27 January 2021. *pyPEAKO* was then trained on these data, leading to the following optimal values for the parameters detailed in Vogl and Radenz (2022): a time averaging window of size 1, a height averaging

window of size 1, a smoothing span of 0.5, a minimum peak width of 0.1 m s$^{-1}$ and a prominence threshold of 0.75 dBZ. After this training step, the algorithm was run on the entire event to label peaks at all (time, range) gates. It was verified that the algorithm yielded similar results as an alternative method where sums of Gaussian-shaped peaks are fitted to the Doppler spectra (Gehring et al., 2022). In addition to the location of each peak, *pyPEAKO* determines its edges; this way, moments ($Z_e$, MDV) can be computed for each identified mode, as well as the SLDR and the signal-to-noise ratio (SNR). Peaks with

a low SNR ($< -15$ dB) are discarded from our analysis. Eventually, for each time and range gate, a number of valid peaks is estimated, for which the radar variables are stored.

### 3.2  Identification of hydrometeor types in multi-modal spectra

To refine the interpretation of multi-modal Doppler spectra, an approach similar to the one of Luke et al. (2021) is implemented. The purpose is to classify the secondary modes when two or more peaks are identified in the spectra. Here and further, the

*primary mode*, sometimes referred to as the *rimer*, denotes the peak with the largest Doppler velocity, while the *secondary modes* are all the slower-falling modes; this distinction between primary and secondary peaks is purely velocity-based and





independent of reflectivity values. To identify the type of particles that cause a Doppler spectral mode, the spectral (S)LDR is highly relevant. As pointed out in e.g., Oue et al. (2015); Luke et al. (2021), high (S)LDR values in zenith-pointing measurements imply either the presence of prolate (needle-like or columnar) crystals, or, when visible only in a restricted altitude

range, of melting particles (Ryzhkov and Zrnic, 2019). Conversely, extremely low, or even below-noise-floor (S)LDR values reflect the presence of particles that are symmetrical with respect to the electromagnetic propagation direction, i.e., "disk-like" in the radar view, such as liquid water droplets or planar crystals (Ryzhkov and Zrnic, 2019). Note, however, that the latter are usually associated with slightly higher (S)LDR values. Other types of snow particles such as aggregate snowflakes or rimed particles may lead to medium-low values of (S)LDR depending on their composition and geometry. By examining not only

(S)LDR but also the other radar variables, additional insight can be gained. For instance, cloud droplets are often identified by their signature in the form of a narrow, low-reflectivity peak with Doppler velocity close to zero (Li and Moisseev, 2019; Li et al., 2021; von Terzi et al., 2022).

The proposed approach aims to combine the information contained in spectral variables ($\mathrm{MDV}_m$, $Z_{e,m}$, $\mathrm{SLDR}_m$ of the secondary peaks, where the subscript "$m$" indicates that the quantities correspond to a single spectral mode) in a comprehensive

manner, to facilitate the visualization and interpretation of spatio-temporal features of Doppler modes.

When at least two peaks are detected, the following decision tree is applied to the *secondary peaks* to classify them into a particle type (we recall that only peaks with $\mathrm{SNR}_m > $ -15 dB are considered):

- $\mathrm{SLDR}_m > $ -20 dB: *columnar crystals*.

- $\mathrm{SLDR}_m < $ -28 dB and $Z_{e,m} < $ -18 dBZ and $\mathrm{MDV}_m > -0.6$ m s$^{-1}$: *cloud liquid water droplets*.

- $\mathrm{SLDR}_m < $ -25 dB and not classified as cloud liquid water: *disk-like particle*, a category which may include planar crystals (pristine or rimed) or large droplets.

- Secondary mode not classified in the prior categories: *other*. This may include, for instance, aggregates or other rimed particles.

The threshold values were chosen based on the literature (e.g., for SLDR: Oue et al., 2018; Luke et al., 2021, for $Z_e$: Kogan

et al., 2005, for MDV: Li and Moisseev, 2019; von Terzi et al., 2022) after adjustments based on a few individual spectra from our case study. In particular, for Doppler velocity, using a stricter threshold led to discarding some profiles that were affected by radial air motion (e.g., downdrafts). The SLDR threshold used to detect columnar crystals is also rather low compared to studies where values up to -16 dB are sometimes used (Oue et al., 2018); this choice was made to improve the spatial consistency of the detection. Note that possible attenuation of W-band reflectivity caused for instance by liquid water cloud layers may affect

minimally the output of this classification, in the identification of cloud liquid droplets vs. disk-like particles. However, the results of the classification show overall little sensitivity to the selected threshold values: only the exact spatial and temporal extent of the regions identified as containing one type of particles are affected by these thresholds, but not per se the existence of these regions, their general behavior, or their approximate location.



Figure 1 shows an example of secondary mode labeling where two main categories are identified: columnar crystals and
cloud liquid droplets. This time step was chosen as it corresponds to an overpass of the aircraft above the ground site and
offers the opportunity to validate the proposed classification. In general, a single spectrogram should not be interpreted as the
trajectory or history of a particle population: the particles in the lower layers do not necessarily originate from the upper layers,
and this can be misleading in heterogeneous or non-stationary systems. In periods with reasonable temporal homogeneity as is
the case here, however, one can still look for signatures of processes in Doppler spectrograms. Alternatively, fall streak tracking
can be implemented (e.g., Kalesse et al., 2016; Pfitzenmaier et al., 2017). This method can also have shortcomings as it relies
on wind profiles estimated from model data or interpolated radiosoundings; it also requires to neglect directional wind shear,
which, in this case study, seemed an overly strong assumption (with the presence of at least one layer of significant directional
shear, not shown).

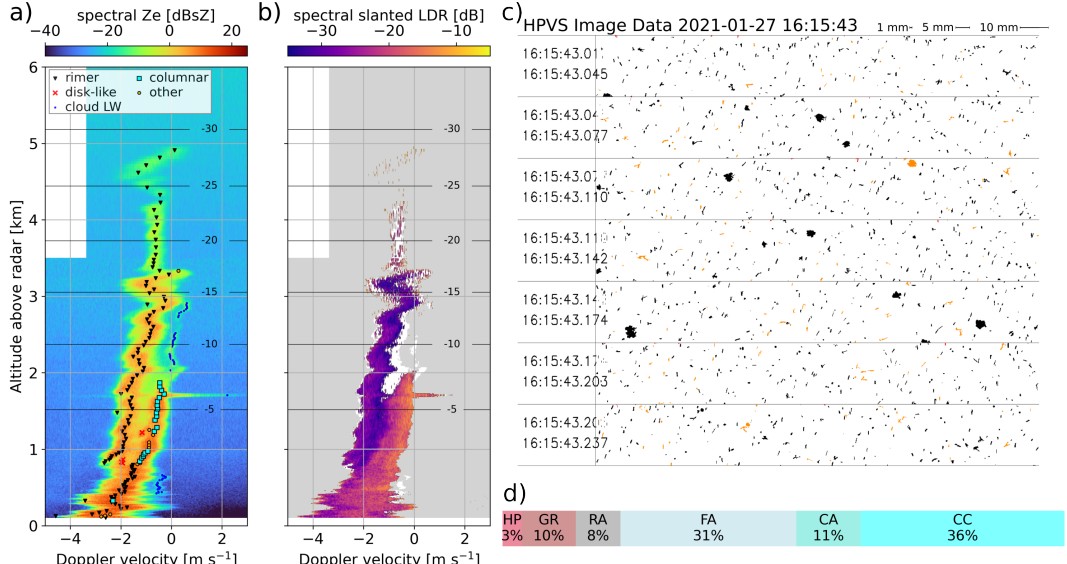

**Figure 1.** (a) Example of Doppler spectrogram where secondary modes are labeled according to the described decision tree. (b) Corresponding SLDR spectrogram; the white area around the spectrograms is where the cross-polar signal is below the noise level, but where the co-polar signal is strong enough that a SLDR higher than -18 dB would be detected; the gray area is where the cross-polar signal is below noise level and where the co-polar signal is too low for a SLDR up to -18 dB to be measurable. In panels (a) and (b), the temperature profile is interpolated from WRF model output. (c) Aircraft HVPS image from the time step where the aircraft overpasses the radar, at an altitude of 1700 m above ground; orange particles are flagged by the built-in algorithm of the probe as possibly shattered, but this does not impact qualitative analyses of the images. (d) Habit classification from PIP images at the same time step (16:15 UTC). HP: hexagonal planar crystals; GR: graupel; RA: rimed aggregates, FA: fragile aggregates, CA: aggregates of columns/needles, CC: columnar crystals (columns/needles).

In the reflectivity spectrogram (Fig. 1a) one can identify the rimer, precipitating from higher regions and progressively
reaching high fall velocities ($|\mathrm{MDV}_{rimer}| > 2 \text{ m s}^{-1}$). Meanwhile, as it accelerates between 3 and 2 km, it coexists with a
population of hydrometeors whose signature is a narrow mode with low reflectivity, negligible fall velocity, and a low—even





below noise level—SLDR (Fig. 1b) : this is a likely signature of SLW, and the fact that the primary mode accelerates at the same time, suggesting riming, supports this interpretation. Below 1.8 km, a secondary mode with much higher reflectivity, spectral width, and most strikingly high SLDR is visible: this would correspond to columnar or needle-like crystals and is labeled as such by our classification routine. Simultaneous aircraft measurements at 1700 m support this reading: in the HVPS images (Fig. 1c), a few large heavily rimed or graupel particles can be seen, as well as numerous columnar particles and aggregates of needles or columns. The independent PIP-based morphological classification (Jaffeux et al., 2022, for particles with a maximum dimension greater than 2 mm,) shown in Fig. 1d also confirms this partitioning, with 18% of rimed particles (graupel and rimed aggregates), 36% of columnar crystals and 42% of aggregates which are either distinctly classified as made of columns and needles, or simply labeled as fragile, which denote weakly bounded crystals).

## 4  Overview of the case study

### 4.1  Synoptic situation

On 27 January 2021, LCDF was located behind a trough directing a strong northwesterly flow over Switzerland (Figure 2b). A warm front associated with a deep low-pressure system over the North Atlantic (Figure 2a) led to stratiform precipitation. At the surface, this translated into an increase of temperatures in two stages, first in the morning of 27 January (06 UTC - 12 UTC), then on 28 January at 06 UTC (see for instance, Appendix Fig. C2). Between these two time frames, surface temperatures were roughly around or slightly above 0°C; snowfall was observed at the ground until 21UTC on 27 January.

### 4.2  Radar timeseries

Height-time plots of WProf reflectivity and mean Doppler velocity are displayed in Fig. 3. Here we point out a few distinct features visible in these time series. A low-level cloud layer persists through the event around 800–1000 m above ground, visible at first (before 10:30) in the $Z_{e,W}$ and MDV fields (panels a and b), then as a persistent layer with multi-modal spectra through which ice particles from higher levels sediment (panel c). Collocated zenith-pointing lidar measurements available between 11 and 12 UTC (not shown) confirm the presence of cloud liquid water droplets in this region, identified as a layer with strong lidar backscatter above which the signal is extinct.

Around a similar altitude, a layer of enhanced reflectivity can sometimes be observed ($\sim$ 12:30, $\sim$ 15:30–16:15 UTC, $\sim$ 19:10–19:50 UTC). This reveals the presence of a partial melting layer related to the onset of the warm front, during which a warm air mass with slightly positive temperatures overlays, then replaces, a cooler air mass with negative temperatures (e.g., Emory et al., 2014). This temperature inversion is confirmed by aircraft measurements of air temperature (Billault-Roux et al., 2023b).

Another noticeable feature that comes across from the radar time series is the presence of multiple—at least three—cloud layers, which are first distinct in the hours before 12 UTC, and then merge in the radar signature as particles precipitate from the higher clouds through the lower ones, in a seeder/feeder configuration. This is particularly visible in the time frame 11:40





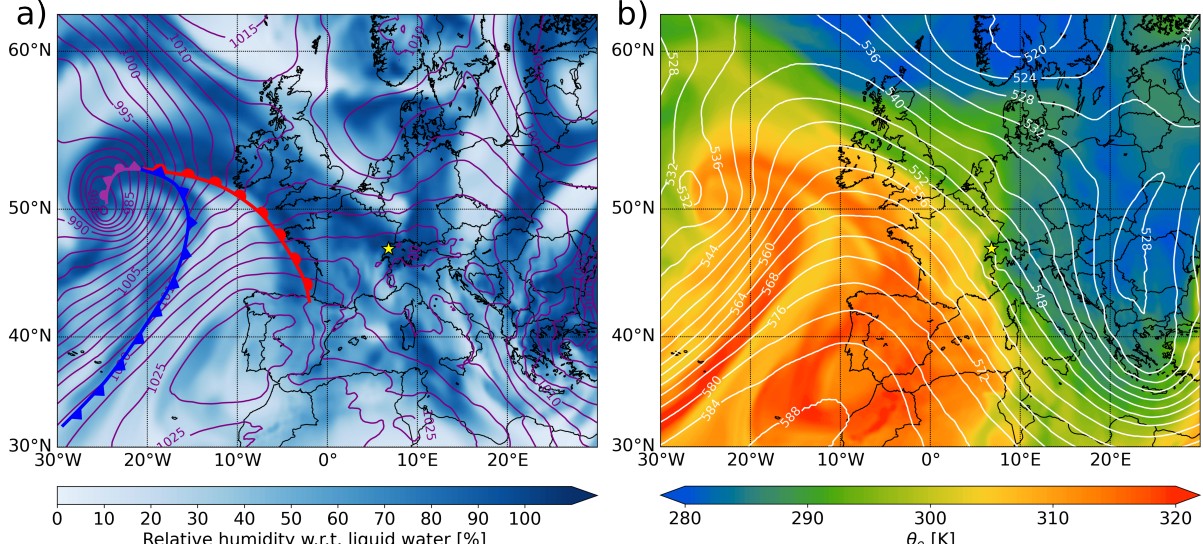

**Figure 2.** Synoptic map on 27 January at 12 UTC from ERA5 data (Hersbach et al., 2020). (a) Relative humidity at 700 hPa (shading) and mean sea level pressure (contours, labels in hectopascal). The blue, red, and purple lines represent the cold, warm, and occluded fronts, respectively (analysis based on 850 hPa temperature, mean sea level pressure, and satellite images). (b): Equivalent potential temperature $\theta_e$ at 850 hPa (shading) and geopotential height at 500 hPa (contours, labels in decameter). The yellow stars indicate the location of LCDF. Adapted from Billault-Roux et al. (2023b).

to 12:05 UTC, between 3 and 5 km above ground: snow particles formed in the overlaying cloud (4-6 km) precipitate above a feeder cloud (extending from 2 to 3 km), which they reach around 11:55 UTC causing a reflectivity enhancement. The en-
hancement at this altitude continues to be observed after this, which leads to believe that the seeding mechanism persists, as external or possibly in-cloud seeding (Proske et al., 2021). This interpretation is reinforced by observations in the following sections.

One of the most striking observations is the persistent Doppler spectral multimodality which has a significant extent in both
height (2 to 3 km) and time (apparently from 14:50 to at least 21:00, assuming that there is a degree of continuity during the time period where data are missing). The rest of the investigation will focus on the multi-modal features during this time frame.

The results of the labeling procedure described in Sect. 3.2 are shown in Fig. 4, focusing on a shorter time frame. In Fig. 4a, (time, range) gates where a secondary population is labeled as one of the four types (*columnar*, *cloud LW*, *disk-like*, *other*)
are visualized as semi-transparent colored layers: this way, the spatial and temporal signatures of the different hydrometeor populations and their coexistence can be analyzed. One noticeable feature in this time series is the lower-level liquid cloud (sometimes labeled as disk-like particles), which was mentioned earlier and corresponds to the pre-existing low-level cloud already visible from ∼08:00 UTC. The presence of liquid water seems however not restricted to this layer, as cloud liquid




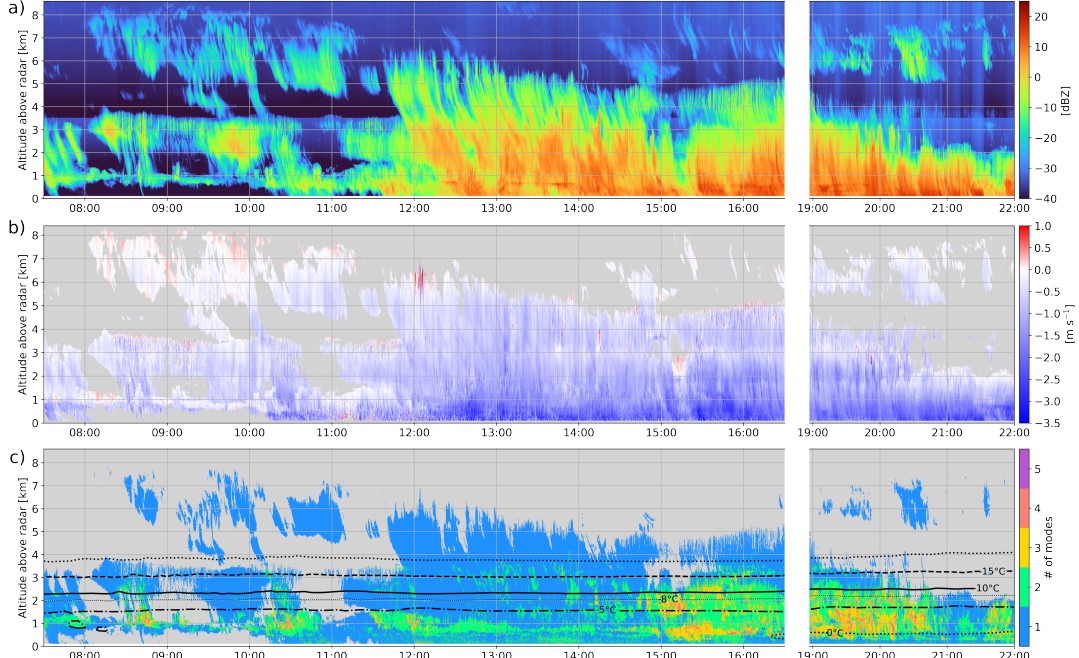

**Figure 3.** Time series of WProf radar moments on 27 January. (a) Reflectivity ($Z_{e,W}$); (b) mean Doppler velocity (MDV), with the sign convention that downward motion is negative; (c) number of modes detected with pyPEAKO (Kalesse et al., 2019) in the Doppler spectra, with overlaid temperature contours (WRF simulations). SNR thresholds are applied in panels (b) and (c), with values of resp. -15 dB and -8 dB. Data collection was interrupted from 16:30 to 18:57 due to a power shortage; the later stage of the event, after 19:00, is included to show the persistence and eventual decay of the cloud system.

is detected between 1.5 and 3 km from ∼14:45 to ∼16:30 UTC, which confirms the existence of a high-level feeder layer.
Another striking observation is the detection of columnar crystals, at first in a restricted altitude range around 1.5 km (∼13:15 to 15:00 UTC), then in most range gates below 1.8 km (15:00 to 20:30 UTC). One can observe other spatially and temporally consistent structures which are labeled as a certain particle type. For instance, a disk-like mode is identified either in restricted altitude ranges (e.g., 15:00 UTC, ∼2 km) or in vertically-extended but short-lasting cells (e.g., 16:20). In what follows, we focus our analysis on specific time frames where different signatures are observed and seem to reveal different microphysical
processes.

## 5   Insights into microphysical processes

From the inspection of Fig. 4, it was decided to focus on the signatures observed during three time frames: 14:50-15:20 UTC, 15:25-15:45 UTC, and 16:05-16:30 UTC. By investigating more precisely the radar and in-situ measurements during those phases, we narrow down possible interpretations for these microphysical fingerprints.





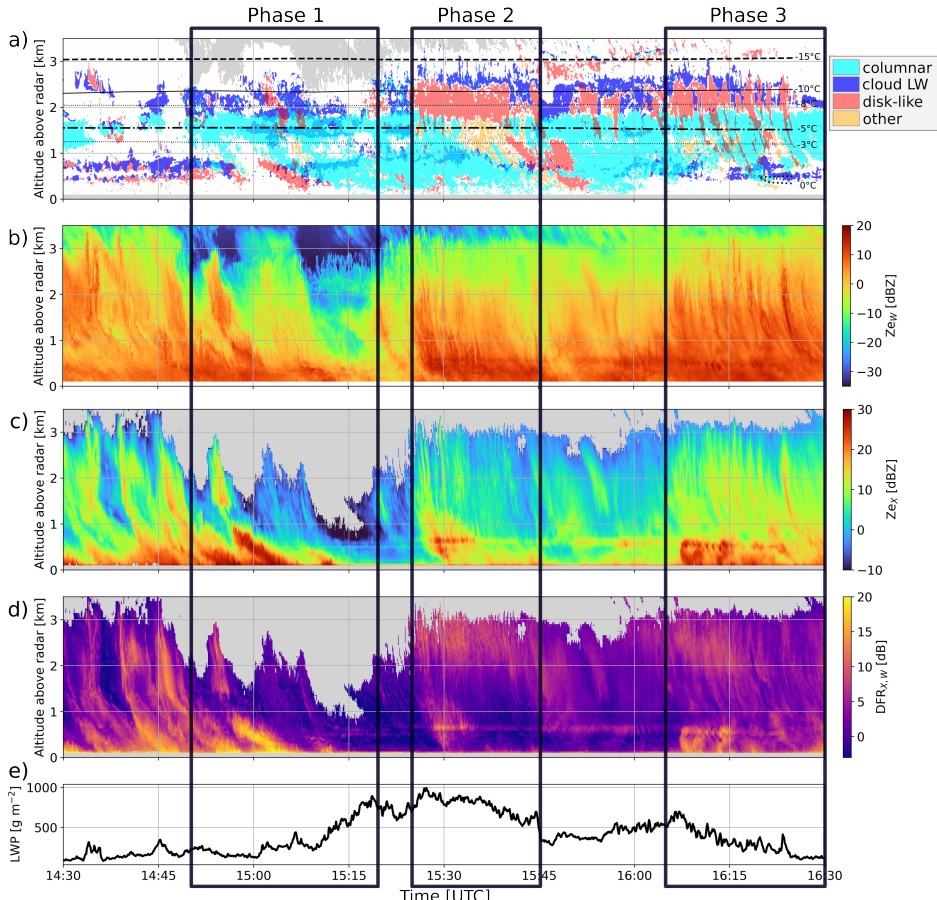

**Figure 4.** Time series covering a subset of the event where the multi-modal features are the most visible. (a) Secondary mode labeling (Sect. 3.2), visualized in the following way: for each of the four types considered, a boolean array is defined which is *true* at (time, range) pixels where a secondary mode is classified as this type; these four layers are then superimposed as semi-transparent layers. Temperature contours from WRF simulations. To reduce the noisiness, a pixel is colored if at least two of its neighbors are labeled with the same type. Height-time plots of (b) $Z_{e,W}$, (c) $Z_{e,X}$ and (d) DFR. (e) LWP time series. The dark boxes indicate the three time frames on which Sect. 5 focuses.

## 5.1 Phase 1, 14:50-15:20: Rime splintering


The first time frame stands out by the presence of a rimer population, a supercooled liquid cloud layer, and a population of columnar crystals, as visible in the time series of Fig. 4a. Figure 5 summarizes these features through the statistics of $Z_{e,W}$ (a) and MDV (b) of each mode during this time frame, together with the number of peaks (c). Panels (d) and (e) illustrate an example of Doppler spectrogram (respectively $sZ_e$ and spectral SLDR) where the most representative features were visible at

once. The range dimension is restrained to the region between 1 and 4 km to focus on the area of interest. From panels (a) and (b) it can be seen that the SLW mode (denoted CLW1) has, as expected, both low reflectivity ($< -20$ dBZ) and low Doppler



velocity (-0.3 to 0.1 m s$^{-1}$). In the upper levels, the primary mode (R1) has a faint signature, with low reflectivity (-30 to -20 dBZ), which decreases from $\sim$ 4 to $\sim$ 3 km; this may reflect sublimation within a drier layer underneath a seeder region, confirmed by the profile of relative humidity with respect to ice simulated with WRF (not shown). $Z_{e,R1}$ (the subscript refers to the hydrometeor population detected) then increases downwards below 3 km (-15°C), while MDV$_{R1}$ increases only slightly ($\sim$ 0.5 m s$^{-1}$). This may correspond to a region planar depositional growth, consistent with the temperature range. When R1 reaches the CLW1 layer around 2.4 km (-10°C), $Z_{e,R1}$ continues to increase and MDV$_{R1}$ accelerates up to $\sim -1.5$ m s$^{-1}$, which indicates riming (Kneifel and Moisseev, 2020), consistent with the interpretation of CLW1 as liquid droplets. Further down, the columnar mode (CC1) is detected, roughly below 2 km. In Fig. 5c, the median of the number of peaks is shown; it illustrates that the three modes (rimer, supercooled droplets, column/needle-like crystals) indeed coexist and do not correspond to different time steps. Collocated in-situ observations are available during an overpass of the ATR42 at 15:05 UTC, 1400 m: 2D-S and HVPS images (Fig. 6a) reveal large graupel particles as well as column- or needle-like crystals, while the CDP-2 confirms the presence of supercooled cloud droplets, with an LWC around 0.1 g m$^{-3}$. These observations support the interpretation of the types of particles corresponding to each mode (heavily rimed R1, pristine CC1, and droplets CLW1).

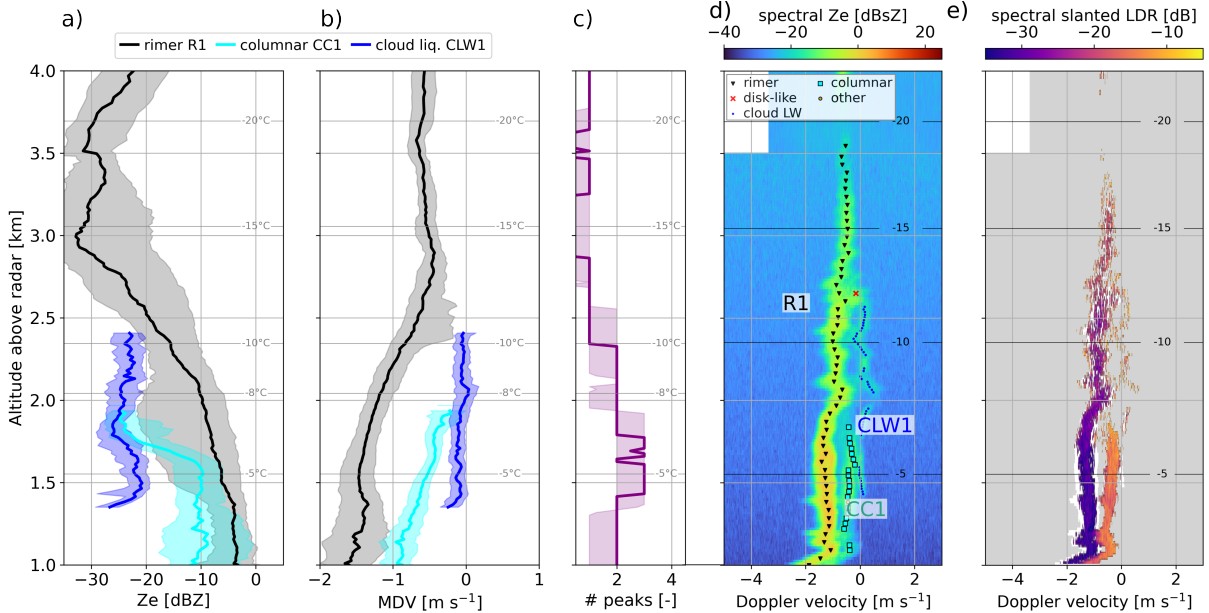

**Figure 5.** (a) $Z_e$; (b) MDV median profiles (with IQR in shaded area) of the different mode types labeled following Sect. 3.2, during the time frame 14:50 - 15:20. Range gates where the modes were detected less than 25% of the time are discarded. (c) Median profile (and IQR) of the number of peaks identified with *pyPEAKO*. (d) (respectively (e)) Example of reflectivity (respectively SLDR) spectrum collected during this time frame (15:01:06), with the modes found and labeled through the methods of Sect. 3. $1\,\mathrm{dBsZ} = 10\log_{10}(1\,\mathrm{mm}^6\,\mathrm{m}^{-3}\,(\mathrm{m\,s}^{-1})^{-1})$ (unit of $sZ_e$). Temperature contours from WRF simulations.




These signatures and the temperature range in which they are observed (slightly above -8°C) suggest that SIP through rime splintering (HM process) may be active: CC1 would result from the splinters produced during the riming of CLW1 onto R1 when temperatures exceed -8°C. It is likely that the HM process is active during most of the event after ~14:30 UTC, as suggested by the persistence of a columnar mode exactly below the -8°C isotherm during this phase (Fig. 4). We chose to focus specifically on the 14:50-15:20 time frame since the signatures in the rest of the event are entangled with other processes, as will be discussed further.

### 5.1.1 Hypothesis of secondary ice production

Radar measurements are not sufficient for an unequivocal identification of SIP occurrence, since those can only be proven through a comparison of ICNC and INP concentrations close to cloud top, obtained with in-situ aerosol measurements (Järvinen et al., 2022) or with Raman lidars (Wieder et al., 2022). In regions where the atmospheric conditions are typically pristine and INP concentrations quite low, the reflectivity of secondary spectral modes can reasonably be used to identify SIP: this is the approach of Luke et al. (2021) where a reflectivity threshold is used (-21 dBsZ), above which the authors consider that the ICNC must be high enough that only ice multiplication processes can account for it. In our case, these thresholds are well exceeded, with the spectral reflectivity of the secondary mode reaching -10 dBsZ in Fig. 5d and exceeding 0 dBsZ at other time steps. However, for a more quantitative approach, we followed the generic method of Li et al. (2021), hereafter LI21, to assess whether we can support the hypothesis that the secondary mode indeed originates in SIP. The goal is to demonstrate that, if this mode were generated through primary ice production, it would require INP concentrations that exceed the expected ones. The steps are as follows (the detail of the equations is provided in Appendix A):

1. Identification of a region as the source of the new ice population: we suppose that it is generated at altitudes slightly above the upper limit of the detected radar signal (here, between 1850 and 2050 m).

2. Simulate the growth by vapor deposition of crystals generated in this region, assuming saturation with respect to liquid water. In this step, particle mass ($m$), size (maximum dimension $D$), and terminal velocity ($v_t$) are modeled using equations of diffusional growth (e.g. Hall and Pruppacher, 1976; Pruppacher and Klett, 2010a). We assess the accuracy of this modeling step by verifying that the obtained terminal velocity $v_t$ agrees with that of CC1 (Appendix Sect. A2). Assuming columnar growth, we obtain (Appendix Fig. A1a–c) a crystal mass of 0.90 to 2.4 $\mu$g corresponding to $D \sim$ 0.12–0.33 mm, at 1.6 km above ground ($v_t \sim$0.29–0.46 m s$^{-1}$). This range of values is obtained by varying the generation height (see step 1) and the aspect ratio of the particles.

3. Estimate the ice water content (IWC) of the secondary mode using literature $Z_e$–IWC relations (e.g., IWC = 0.137 $z_e^{0.643}$ after Liu and Illingworth, 2000, with $z_e$ in mm$^6$m$^{-3}$ [such that $Z_e = 10 \log_{10}(z_e)$] and IWC in g m$^{-3}$). Using the 25% and 75% quantiles of the $Z_e$ profile (-16 to -8 dBZ), this gives an IWC of 0.012 to 0.042 g m$^{-3}$ at 1.6 km. Similar results are obtained with the relations of Aydin and Tang (1997); Boudala et al. (2006). Note that such $Z_e$–IWC relations are, however, associated with rather high uncertainty (e.g., -50% to +100% errors reported in Liu and Illingworth, 2000)



4. A rough estimate of the resulting ICNC is then obtained as ICNC $= \frac{IWC}{m}$, i.e., here ICNC $\sim$ 7–50 L$^{-1}$ at 1.6 km, using the IQR of IWC and the mass estimate obtained earlier.

5. This estimate is compared to typical INP concentrations at the temperature range where the ice particles were assumed to be generated (here, -8 to -10°C). For this, statistics of INP concentrations measured at the high altitude Jungfraujoch (JFJ) measurement site (3580 m asl, approximately 100 km southeast of LCDF) in free tropospheric conditions are taken from Conen et al. (2022). During two years of measurements, Conen et al. (2022) observed concentrations of active INPs at -10°C and -15°C ranging from $1.0 \times 10^{-3}$ L$^{-1}$ to $1.6 \times 10^{-2}$ L$^{-1}$. While no INP measurements are directly available for the event of interest, measurements of the total aerosol number concentration indicate a low aerosol loading on this day (below the lower 10% quantile of the 2020-2021 winter, compiled from condensation particle counter data available through Tørseth et al., 2012, at http://ebas-data.nilu.no/, last access: 7 March 2023): the concentration of active INP on 27 January are thus unlikely to be significantly outside of the statistical bounds of Conen et al. (2022). Another estimate of INP concentrations may be derived from the temperature-dependent relation mentioned in DeMott et al. (2010), which gives values of 0.3 to 0.4 L$^{-1}$ at -8 to -10°C. As underlined by DeMott et al. (2010), this relation has a large uncertainty, and is presumably less trustworthy than the INP statistics at JFJ.

The above approach gives ICNC estimates higher by one to four orders of magnitude compared to expected INP concentrations, which supports the SIP hypothesis. While these estimates are valuable, they are prone to a quite high error as several hypotheses are involved in each step of the method, such as where the ice particles are generated, the mass-dimensional relations used, geometrical description, ventilation coefficients, to list a few (see Appendix A). We can also note that possible riming of the crystals (after they have grown to a sufficient size) would not be adequately modeled by this approach, which considers exclusively depositional growth. All these hypotheses inevitably contribute to an uncertainty propagation which it is both challenging to quantify and to reduce. Without further information on INP concentrations during this specific event, it remains difficult to make strict assertions on the occurrence of SIP through the HM process, although it appears as a reasonable hypothesis in view of the observed signatures and results of the LI21 method.

## 5.2 Phase 2, 15:25-15:45: New ice production in high-LWC region

From 15:25 to 15:45 UTC, a mode labeled as "disk-like" (DL2) is persistently identified between 1.8 and 2.2 km. Figure 4, from a time series perspective, and Fig. 7a–b from a statistical summary perspective, suggest that DL2 is below a layer of SLW droplets, and above a population with higher SLDR (labeled either as "columnar" or as "other"). Following the rationale of Sect. 3.2, the low SLDR$_{DL2}$ ($< -25$ dB) together with relatively high reflectivity ($Z_{e,DL2} >$-10 dBZ, Fig. 7) and MDV$_{DL2}$ (down to -0.5 m s$^{-1}$ ) of this peak suggests that it is composed of either planar crystals or larger supercooled droplets (drizzle). Fully resolving this question is challenging, but a few steps can be achieved to improve the understanding of these microphysical signatures.





**Figure 6.** HVPS and 2D-S images for the three time frames: (a) 15:05; (b) 15:40; and (c) 16:20. Panels a and b correspond to overpasses over the radar. The scale of the HVPS images is indicated at the top. The vertical bar in the 2D-S images corresponds to 1.28 mm. The orange particles were flagged by the HVPS built-in software as possibly affected by shattering within the probe. Circled are examples of particles discussed in the text: liquid droplets/drops (blue), heavily rimed particles (purple), "lollipop"-shaped particles (red), (fragments of) pristine dendritic crystals (green).

### 5.2.1 Presence of liquid droplets

Several independent observations point to the presence of liquid water in this region, thus suggesting that the secondary mode DL2 is at least partly caused by liquid water droplets. The first element is the increase in fall velocity of the rimer mode (R2), from 1 to 2 m s$^{-1}$ between 2.5 and 2 km. This increase already begins in the region of the cloud droplet mode (2.5 - 2.8 km), but continues below. Fall velocities of this order (e.g., larger than 1.5 m s$^{-1}$) are typically used to identify rimed particles





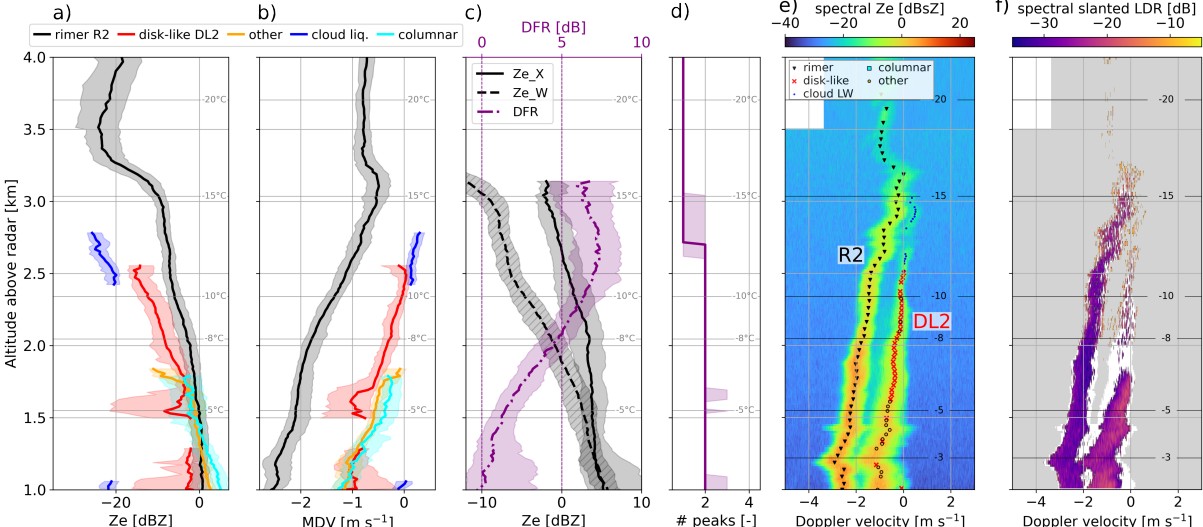

**Figure 7.** (a) $Z_e$, (b) MDV median profiles (with IQR in shaded area) of the different mode types labeled following Sect. 3.2, during the time frame 15:25 – 15:45 UTC. Range gates where the modes were detected less than 25% of the time are discarded. (c) Black lines (values on bottom axis): median profiles of $Z_{e,X}$ (full) and $Z_{e,W}$ (dashed); purple line, with values on upper x-axis: median DFR profile (and IQR). (d) Median profile (and IQR) of the number of peaks identified with *pyPEAKO*. (e) (respectively (f)) Example of reflectivity (respectively SLDR) spectrum collected during this time frame (15:36:28 UTC), with the modes found and labeled through the methods of Sect. 3. Temperature contours from WRF simulations.

(Kneifel and Moisseev, 2020) and consequently suggest the presence of supercooled droplets.

Secondly, the LWP time series (Fig. 4 e) reaches remarkably large ($> 800$ g m$^{-2}$) values during this time frame. While the
365 LWP retrieval does not inform on the altitude of the liquid cloud layers, it does confirm the presence of SLW in this period.

Lastly, we can leverage the collocated X-band measurements, shown in Fig. 4 with both $Z_{e,X}$ (panel c) and DFR (panel d). DFR is often used in radar-based studies of snowfall microphysics as a proxy for particle size, but can also serve as a way to quantify W-band attenuation (Hogan et al., 2005; Tridon et al., 2020). In the time frame on which this subsection focuses, high DFR ($>10$ dB) coinciding with relatively low $Z_{e,X}$ ($\sim 5$ dBZ) are observed up to echo top, while low DFR values are expected
370 in such regions where crystals are usually in an early growth phase. This suggests that the enhanced DFR is not related to the presence of large particles but rather to an abrupt attenuation of the W-band signal, caused by a layer with significant LWC. Panel c of Fig. 7 illustrates the median DFR profile between 15:25 and 15:45 and confirms what was observed in the time series, with a DFR that increases in the region where DL2 is present, and does not decrease to 0 dB near echo top, suggesting that the increase is related to W-band attenuation. As DFR values are low in the bottom part of the profile, little to no attenuation is
375 expected in this region, meaning that most of the SLW droplets would be in the region of DL2.



These elements are evidence that a population of liquid water droplets is at least partly responsible for the DL2 signature. To quantify, or at least constrain, the properties of these droplets in this region, we can combine the information from 1/ $Z_{e,DL2}$ and $\text{MDV}_{DL2}$, both of which would be related to the size of the drops (assuming DL2 consists only of liquid water), and 2/ the attenuation caused by DL2, which reflects the total LWC (if all droplets are small enough to be within the Rayleigh scattering approximation). We use the radiative transfer model PAMTRA (Mech et al., 2020) to simulate the attenuation and reflectivity of a cloud/drizzle population as a function of the LWC and the drop size distribution (see detail of the simulations in Appendix B). We then rely on the measurements of DL2 as constraints on reflectivity (between -15 and -2 dBZ), specific attenuation (between 4 and 6 dB km$^{-1}$) and mean Doppler velocity ($0.15 \, \text{m s}^{-1} < |\text{MDV}| < 0.5 \, \text{m s}^{-1}$). With a simple look-up table approach, this translates into bounds on LWC and median volume diameter ($D_v$, such that half of the volume of water is contained in droplets smaller than $D_v$): $0.9 \, \text{g m}^{-3} < \text{LWC} < 1.4 \, \text{g m}^{-3}$, and $35 \, \mu\text{m} < D_v < 70 \, \mu\text{m}$. These bounds are quite rough, in particular since we considered that only liquid droplets (i.e., no ice crystals) contributed to $Z_{e,DL2}$. They do, nonetheless, highlight the presence of significant LWC and likely of large ($> 50 \, \mu$m) droplets, although this is not sufficient to claim that DL2 consists solely of liquid drops.

### 5.2.2 New ice formation

In fact, some signs suggest that the "disk-like" DL2 mode may also contain non-liquid particles. Although they are relatively large and with non-negligible fall velocity, the liquid drops do not precipitate to the ground, or else the attenuation would occur at lower altitudes: hence the liquid content is somehow depleted. Riming is likely not the only process through which this happens, as the DL2 mode does not vanish away in the lower regions, but slowly evolves into a higher-SLDR ($> -25$ dB) mode (pointing to aggregate or column-like snow particles). This implies that some ice crystals are formed within this "disk-like" region, and coexist with liquid droplets. To support this, in Fig. 8, we look at an individual time step instead of global statistics. There, DFR increases only toward the upper part of DL2: this suggests that, at this time step, the LWC of the lower region is only moderate, and that SLW drops are not the only population contributing to the reflectivity.

These elements point to the production of non-columnar ice crystals between 1.8 and 2.5 km, i.e., -7 to -12°C, through heterogeneous freezing of the cloud droplets and/or by SIP. Among the supposedly prominent SIP mechanisms, rime splintering would be unlikely because of the relatively cold temperatures at the top of DL2; given that drizzle-size drops ($> 50 \, \mu$m) might be present, droplet shattering appears as a possible mechanism, although collisional break-up cannot be excluded altogether.

Unfortunately, no aircraft overpasses took place directly in this region (1500 m – 2500 m), but one overpass at 15:40 at 1100 m is still instructive (Fig. 6 b). In the 2D-S images, one can identify (red frames) columnar crystals that grew onto rather large spherical or semi-spherical particles. These are likely frozen drops or fragments of frozen drops, which were formed within the DL2 region: they then served as germs for crystal growth by vapor deposition, with temperatures just above -10°C favoring columnar growth. Similar structures were reported by Korolev et al. (2020) in conditions where droplet shattering was suspected. Such shapes could also explain the only moderately high SLDR values measured in the region of columnar growth. In addition to these "lollipop" shapes, a few images of large drops are collected by the 2D-S (blue frames in Fig. 6 b); a precise





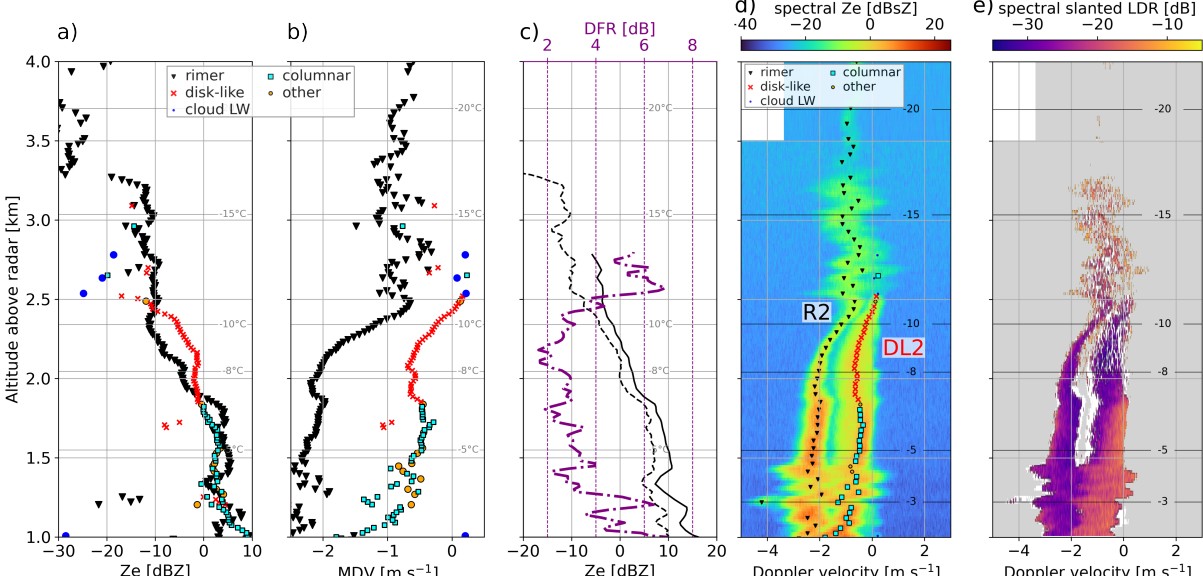

**Figure 8.** (a) $Z_e$, (b) MDV profiles of the different mode types labeled following Sect. 3.2, at 15:27:53. (c) Black lines (values on bottom axis): profiles of $Z_{e,X}$ (full) and $Z_{e,W}$ (dashed); purple line, with values on upper x-axis: DFR profile. (d) (resp. (e)) Reflectivity (resp. SLDR) spectrum collected at the same time step, with the modes found and labeled through the methods of Sect. 3. Temperature contours from WRF simulations.

estimate of droplet size is difficult to make based on these images due to possible diffraction by out-of-focus drops (Korolev, 2007; Vaillant De Guélis et al., 2019). These in-situ images are compatible with the analysis, that drizzle-size liquid droplets are involved in the formation of new ice particles in the region of the DL2 mode. They also suggest that collisional breakup would not be the dominant process, as no signs of fragments of crystals are apparent (Ramelli et al., 2021).


Overall, the above elements suggest droplet shattering as a possibly active mechanism given 1/ the high LWC reflected by W-band attenuation (detected through high DFR values), 2/ the presence of large droplets inferred from the enhanced reflectivity and increase in Doppler velocity of the secondary mode, 3/ the signs of ice formation within this DL2 mode, 4/ the in-situ observations which reveal (fragments of) frozen droplets upon which crystalline growth occurred, 5/ the temperature range,

which is compatible with droplet shattering but not HM.

However, because the liquid droplet and new ice signatures are intertwined in DL2, it is very challenging to disentangle them further to reliably narrow down the dominant microphysical process—primary or secondary ice production. We employ the LI21 method as in Sect. 5.1.1, to get a rough estimate of the potential discrepancy between INP and ICNC. Assuming

the formation of ice particles around 2450 to 2500 m and subsequent growth by vapor deposition and sedimentation (see Appendix A, Fig. A1d–f), at an altitude of 2200 m the particles would have grown to a mass of $2.9 - 26$ $\mu$g (maximum





dimension 0.37 to 2.5 mm, terminal velocity 0.31 to 0.42 m s$^{-1}$). The IWC retrieved from $Z_{e,DL2}$ values (-15 to -7.7 dBZ, assuming this time that $Z_{e,DL2}$ is dominated by ice crystals) would range from .019 to 0.054 g m$^{-3}$, which in the end leads to an estimation of ICNC in the order of 0.7 to 20 L$^{-1}$. The spread is significant due to the uncertainties in modeling particle habit

in this temperature range (-12 to -9°C), where the dominant growth mode shifts from planar to columnar; for this reason, both habits were considered in the simulations, leading to a large spread in the modeled masses and sizes. The retrieved ICNC are here again above the typical active INP concentrations in this temperature range measured at JFJ (Conen et al., 2022), although the discrepancy is slightly less obvious than in the first case (still one to four orders of magnitude higher than JFJ statistics, but within zeros to two orders of magnitude compared to the temperature-based estimate). We highlight that the reflectivity

values used here are affected by significant attenuation; in that sense, the ICNC estimates that we give are rather conservative. If $Z_e$ values are corrected from 4 dB of attenuation (see Sect. 5.2.1), slightly higher ICNC are obtained ranging from 1 to 30 $L^{-1}$. However, it was now assumed that the $Z_{e,DL2}$ values are dominated by ice crystals rather than liquid droplets (by contrast with the previous paragraph): overall, these results do not allow for a clear-cut demonstration of SIP occurrence. It is possible that droplet freezing (upon INP immersion, or collision with ice crystals), and not necessarily shattering, is at least

partly responsible for DL2.

If droplet shattering were taking place, it might, in any case, not be highly efficient in the production of secondary ice crystals. Indeed, Korolev and Leisner (2020) and studies mentioned therein (e.g., Lauber et al., 2018) suggest that the efficiency of droplet shattering upon freezing increases as the supercooled drops become larger. Our analysis, although it does point to the possible presence of droplets with a diameter sufficient to cause shattering of the droplets upon freezing, does not provide

evidence that very large drops (e.g., $> 300$ $\mu$m) are present. In these conditions, droplet shattering might only be moderately efficient in the sense that only a few fragments would be generated per freezing drop, leading to a modest enhancement of ICNC through SIP, consistent with the retrieved estimates.

### 5.2.3   Formation of large droplets

The seeder-feeder configuration, involving a SLW feeding cloud layer with top around 3 km, seems to be an essential driver

of the microphysical signatures discussed up to now. Even though the persistence of mixed-phase systems is frequently acknowledged in the literature (Lohmann et al., 2016), it is instructive to investigate the mechanisms behind the maintenance of the supersaturation over liquid water in the feeder cloud, and the occasional formation of drizzle-sized drops as discussed above. For this purpose, the WRF simulations of the event provide relevant insights into the origin of the air masses and the supercooled liquid clouds. A cross-section along the main wind direction (135°) at 15:20 is shown in Fig. 9, together with a

time series of simulated ice and liquid water content. One first observation from the time series is its rather good agreement with the radar measurements and some of the baseline interpretations that were proposed (Sect. 4.2): the presence of a warm nose as a sign of the warm front onset, visible in the converging contours of potential temperatures slightly below 1 km (especially clear before 12 UTC), and the corresponding low-level liquid water cloud which persists around 1 km with a slowly decreasing altitude (cf. Sect. 4.2). More specifically, the time series also indicates the presence of a higher-level supercooled

cloud (with a top at 3 km), which act as a feeding layer for ice crystals precipitating from above. This SLW cloud is present in




the WRF simulation starting around 12:00 UTC and decaying in strength after 16:00 UTC. LWC is highest around 15:00 UTC with values exceeding 0.5 g m$^{-3}$ around 2.5 km, which is compatible with the radar-based interpretations conducted above (although with a slight temporal shift).

The cross-section (Fig. 9b–d) helps us understand the origin of the enhanced LWC. It appears related to a combination of
large-scale moisture supply—associated with the warm front extending from the Northern Atlantic—with a local enhancement due to orographic lifting over the Jura, efficient since the northwesterly flow is approximately orthogonal to the mountain range. This is confirmed by the vertical velocity field (Fig. 9c), with updrafts visible in upsloping areas, and in the cross-section of the liquid water mixing ratio (Fig. 9b) which is enhanced above the ridge of the Jura around 3.5 km above sea level (2.5 km above ground).

In Fig. 9d, the moist Richardson number (Ri, which is the ratio of buoyancy to wind shear, is used to characterize atmospheric stability, Hogan et al., 2002) at this location indicates a slight dynamic instability (Ri $\sim$ 0.6) near cloud top; this low-Ri region seems to cover a large spatial extent and roughly corresponds to the upper cap of the mesoscale cloud (i.e., windward of the Jura). While these values are not strictly speaking descriptive of a strong dynamic instability (for which a typical threshold is Ri $\leq$ 0.25), they suggest that shear-driven turbulence and/or isobaric mixing may be present and contribute to sustaining
the LWC of the cloud, and possibly inducing the formation of larger droplets (Korolev and Isaac, 2000; Pobanz et al., 1994; Grabowski and Abade, 2017). Overall, the WRF analysis shows that the saturation over liquid water and formation of cloud droplets is triggered by a combination of orographic and frontal lifting, with a possible contribution from shear-induced mixing that favors the formation of larger drops between 15:00 and 15:30, as modeled in WRF and observed in our analysis.

### 5.3 Phase 3, 16:05 - 16:30 UTC: New ice production in turbulent regions

From 16:05 to 16:30, another type of process appears to be happening. In Fig. 4, instead of being confined to a fixed altitude range like DL2, the mode labeled as "disk-like" during this time (DL3) seems to be generated at distinct time steps and at specific heights (between 2.5 and 3 km), and then precipitate to lower altitudes. Such spatio-temporal structures are also visible in the later stage of the event between 19:00 and 20:00 UTC. This creates *fall streak* structures, which can be seen in both the classification and the reflectivity time series of Fig. 4. As DL3 precipitates, it coexists with other modes (e.g., columnar
crystals or liquid cloud droplets) while remaining well separated from these. In the supplementary material, a video is included showing the evolution of the Doppler spectra during these fall streak time steps[1]: it clearly illustrates that DL3 is generated in a region of atmospheric turbulence and updrafts; its formation stops when the turbulence and updraft cease, and the hydrometeor population that was formed then settles downwards.

This is summarized through the statistics and the sample spectrum shown in Fig. 10. There, the statistics are computed on
the entire time frame (16:05 to 16:30), except for DL3 from which we specifically extracted the fall streak patterns, identified as regions when $Z_{e,DL3} > -10$ dBZ. To identify turbulent regions, an estimate of the turbulence eddy dissipation rate (EDR) was derived following Shupe et al. (2008), which combines the variance of the MDV (here, of the *rimer* mode, MDV$_{R3}$) with information on horizontal wind and wind shear (here, from WRF simulations). Fig. 10b illustrates that DL3 is detected just

---

[1]For comparison, similar animations are also included for the other two phases





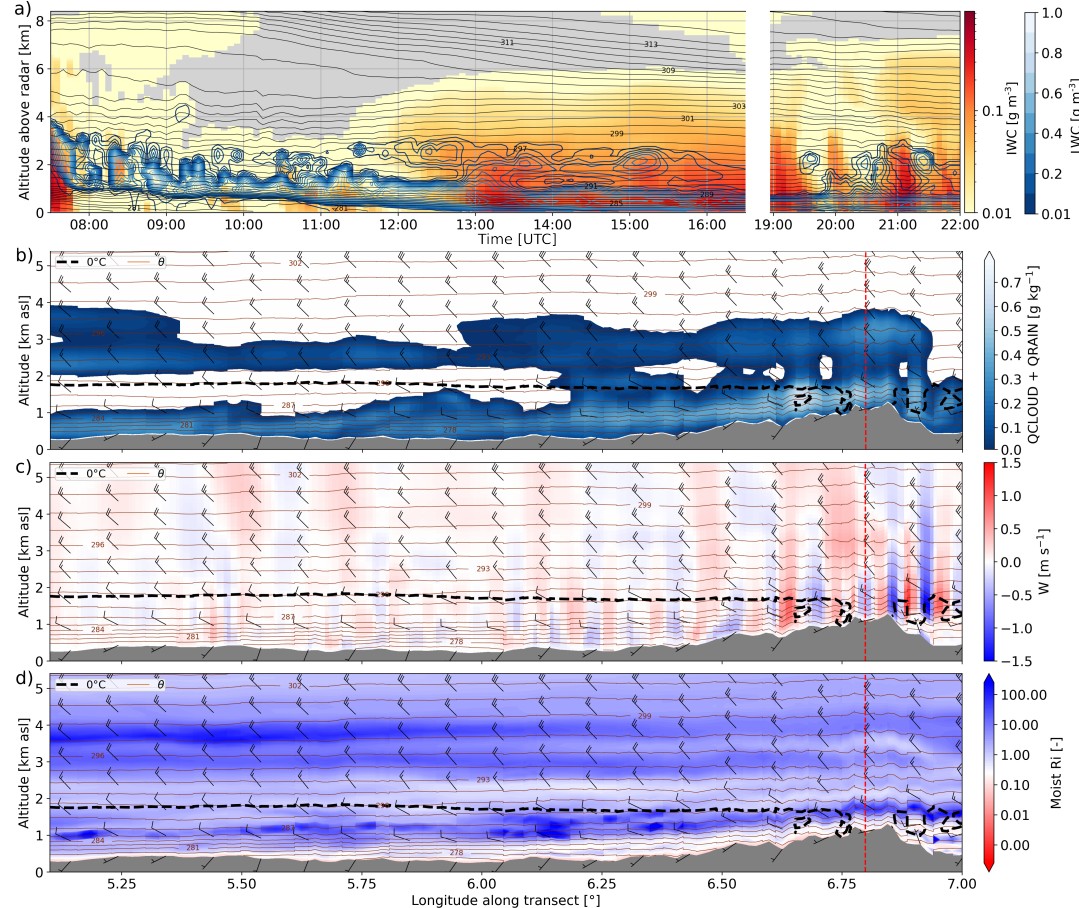

**Figure 9.** (a) Time series of IWC simulated in WRF over La Chaux-de-Fonds (includes ice, snow, and graupel), with LWC in blue contours; (b) (resp. (c), (d)) 15:15 UTC cross-section in direction of the main wind (135 °) with cloud and rain content (resp. vertical wind, moist Richardson number). In all panels, the brown contours indicate the potential temperature; in (b), (c), (d) wind barbs indicate wind speed and direction following standard conventions (in knots) and the black dashed line corresponds to the 0°C isotherm. The vertical dashed red line indicates the location of La Chaux-de-Fonds.

below a region of updraft (seen in a reduction of the rimer MDV) and turbulence (visible in the EDR), between 2.8 and 3.1 km.
In the upper region of DL3 (2.7 km), the mode sometimes coexists with SLW droplets, while lower down it is present along with columnar crystals (Fig. 10a, b, d). While these are not the main focus of this subsection, we can hypothesize that they are formed through rime splintering at temperatures warmer than -8°C, similar to Sect. 5.1.

In terms of radar variables, DL3 combines low $SLDR_{DL3}$ values (< -25 dB) with relatively high $Z_{e,DL3}$ (up to 5 dBZ when looking at individual fall streaks), and $MDV_{DL3}$ around -0.5 to -1 m s$^{-1}$. This suggests that it is composed of planar
ice crystals (or such low-depolarization ice particles) rather than liquid droplets, which would be expected, for instance, to have larger fall velocities for this level of reflectivity (e.g., $Z_e$-V relations for identification of drizzle Luke et al., 2021, and



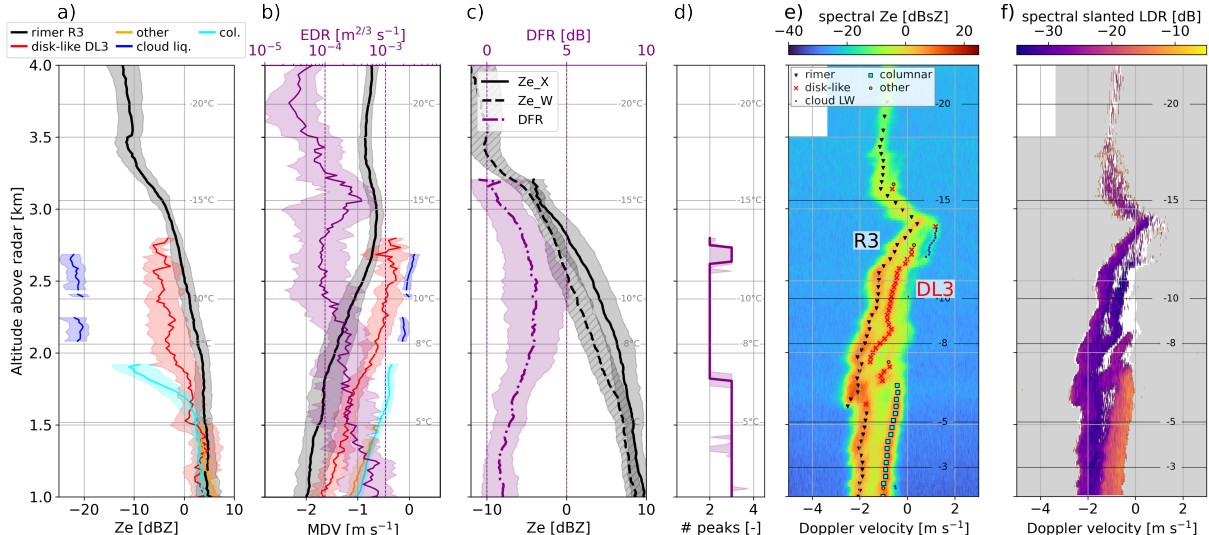

**Figure 10.** (a) $Z_e$ median profiles (with IQR in shaded area) of the different mode types labeled following Sect. 3.2, during the time frame 16:05 - 16:30. Range gates where the modes were detected less than 25% of the time are discarded. (b) Same with MDV (on the bottom x-axis); the turbulent EDR estimated from the rimer mode (Shupe et al., 2008) is shown with the purple line (median and IQR). (c) Black lines (values on bottom axis): median profiles of $Z_{e,X}$ (full) and $Z_{e,W}$ (dashed); purple line, with values on upper x-axis: median DFR profile (and IQR). (d) Median profile (and IQR) of the number of peaks identified with *pyPEAKO*. (e) (respectively (f)) Example of reflectivity (respectively SLDR) spectrum collected during this time frame (16:18:08), with the modes found and labeled through the methods of Sect. 3. Temperature contours from WRF simulations.

their Supplementary Material). The temperature range in the region where this mode is formed (-15 to -12°C) is compatible with planar growth of crystals by vapor deposition. It is worth noting that the DL3 signature is, however, different from the ones typically observed in the dendritic growth layer, in which a small updraft, an increase in reflectivity, and a persistent

spectral bimodality are often reported (von Terzi et al., 2022), and which is occasionally observed during this case study (see for instance, Fig. 3, around -15°C between 15:00 and 16:30, or the spectrogram in Fig. 10 around 3.1 km). By contrast, DL3 is generated in stronger and more localized updrafts (e.g., 2.8 km in Fig. 10e).

An unambiguous identification of the microphysical process(es) leading to the formation of this mode is once again difficult.

SIP is possibly responsible for DL3: the high reflectivity of the new ice mode, only a few range gates below it is formed, indicates a relatively high concentration of ice crystals which would exceed typical values of INP concentrations in this temperature range. This concurs with previous observations of ice multiplication occurring within generating cells leading to fall streak structures (Ramelli et al., 2021). With the LI21 approach, we focus here on a single DL3 fall streak (16:17-16:20) and consider that particles are formed between 2.7 and 2.85 km (see Appendix A Fig. A1g–i). At a height of 2.5 km, they would

have grown (assuming plate-like or dendritic crystals) to a mass of 9.5 to 35 $\mu$g ($D \sim 0.82$ to 2.8 mm); the IWC estimate from



$Z_{e,DL3}$ values (-4.1 to 1.4 dBZ) ranges from 0.074 to 0.17 g m$^{-3}$ and the resulting ICNC = 2 to 20 L$^{-1}$ once again exceeds the typical active INP concentration (1.0–16×10$^{-3}$ L$^{-1}$ following Conen et al., 2022; 0.5 to 0.6 L$^{-1}$ with the temperature-only relation of DeMott et al. (2010)). As in phase 2, the attenuated reflectivity values are used here, so this would rather underestimate the true ICNC. Similar to the previous sections, these values are subject to uncertainty and should be taken with care, but

nonetheless, support the hypothesis that DL3 originates in SIP.

The updrafts and turbulence which contribute to the formation of DL3 also generate SLW droplets: this is seen, for instance, in Fig. 10e, and in the LWP time series (Fig. 4e) where peaks in LWP occur when the DL3 cells/fall streaks are formed. However, the LWC in this region does not cause significant W-band attenuation like was observed in Sect. 5.2.1 and must

therefore be lower. This is especially true when looking at the end of the time frame of interest, after 16:15 in Fig. 4: there is then no DFR increase toward cloud top. Additionally, when the liquid cloud droplets generated by these updrafts are visible as a distinct mode—which is not always the case, since strong turbulence can broaden the spectra to a point where several peaks are merged into one—like in Fig. 10e, it is rather narrow and has a low reflectivity ($\sim$ -20 dBZ), which is rather a sign of small cloud droplets than of drizzle-size drops. With these elements in mind, droplet shattering upon freezing does not seem the most

likely process for the DL3 signatures.

On the contrary, ice multiplication through collisional breakup might be a plausible explanation. In the turbulent updraft region, supercooled droplets may form, onto which the primary population can start riming; meanwhile, in these turbulent eddies, collisions of these newly rimed particles would be favored (Pruppacher and Klett, 2010b; Sheikh et al., 2022) either with one

another or with the still pristine ones (Phillips et al., 2017b), leading to the formation of DL3 particles. These fragments would subsequently grow by vapor deposition (efficient because of the supersaturated conditions), by aggregation, and/or eventually by riming if they reach large enough sizes ($\sim$ 100 $\mu$m, e.g., Pruppacher and Klett, 2010b). R3 and DL3 would then separate in the Doppler spectra below the turbulent region due to their different settling velocities (e.g. Ramelli et al., 2021).

The ATR42, unfortunately, did not overpass the radars at a timestep when DL3 fall streaks are observed, and we must therefore make cautious interpretations of the in-situ observations during this time frame. HVPS images at 16:20 at 1700 m (Fig. 6c) reveal a population of slightly rimed particles, together with a few still pristine dendrites and fragments of dendrites, a clear sign to invoke the presence of the collisional break-up mechanism. The latter two might correspond to the DL3 population (either as pristine dendrites that grew onto small fragments, or directly as fragments generated during break-up), and thus

endorse the proposed interpretation, also considering that there are no signs of shattered drops. Yet, we underline again that the link between the in-situ and radar observations remains hypothetical, as they are not collocated.



## 6 Summary and conclusion

In this work, we investigated snowfall microphysical processes during the passage of a warm front in the Swiss Jura Mountains, involving a multi-layer mixed-phase cloud system. The analyses were primarily based on the measurements of a W-band
spectral profiler, together with in-situ observations from the ATR42 aircraft which performed overpasses above the ground site, as well as LWP and dual-frequency radar measurements (X- and W-band) to quantify atmospheric liquid water. Multi-peak Doppler spectra were observed for several hours and over several kilometers in height above ground, suggesting the occurrence of a number of microphysical processes involving different hydrometeor populations. We proposed a labeling method that allows for the systematic identification of certain hydrometeor types in these Doppler spectra, making use of the spectral
polarimetric variables. Specifically, supercooled cloud droplets were distinguished from columnar crystals, and from disk-like particles that may include drizzle-size drops or planar crystals. This way, it became apparent that various hydrometeor habits were causing the multi-modality at different heights and time steps of the event.

Three time periods stood out, during which the multi-modality was attributed to distinct processes. In each case, secondary
ice production appeared as a likely cause for the formation of the new spectral peak(s). Looking into the Doppler spectra in more detail, we proposed interpretations of the mechanisms during the different time frames. The presence of a seeder-feeder configuration seemed to play an essential role in the microphysics of this event. During the three phases, ice crystals precipitated through a SLW layer around 2-3 km above ground, whose presence was identified through cloud radar Doppler spectra, confirmed by WRF simulations and consistent with LWP estimates. In the first phase, the interaction between the rimer and the
SLW cloud led to the formation of columnar ice particles at temperatures warmer than -8°C, pointing to HM rime splintering (Fig. 11a), while no new ice formation was detected at colder temperatures during this time frame. The second phase (Fig. 11b) was associated with an enhancement of the SLW layer, in terms of both LWC and droplet sizes, with the formation of drizzle-size drops. In these conditions, droplet freezing—either through INP immersion, or upon collision of a drop with an ice crystal—and/or shattering may have been active, and involved in the emergence of a new spectral mode below $-10°C$. Lastly,
new ice formation was observed at cold temperatures ($\lesssim -12°C$), toward the top of the SLW cloud region, in localized generating cells associated with strong updrafts and turbulence; these ingredients would favor the riming of the seeding population, and SIP through ice-ice collisions between these newly rimed particles (Fig.11c). The resulting signatures are rather complex, and were narrowed down by combining dual-frequency, Doppler spectral radar measurements, and in-situ images.

A simple modeling method following Li et al. (2021) (Sect. 5.1.1, Appendix A) was implemented for each of these phases and suggested that primary ice production through heterogeneous nucleation could not explain alone these signatures (especially phases 1 and 3), with ICNC estimates exceeding expected INP concentrations by one to four orders of magnitude, hence supporting the SIP hypothesis. This discrepancy is in agreement with previous observations in orographic clouds, especially under seeder-feeder configurations (e.g., Lloyd et al., 2015; Georgakaki et al., 2022). Uncertainties related to this modeling are,
however, substantial: it involves assumptions on ice microphysical properties such as geometry, mass-dimensional or velocity-





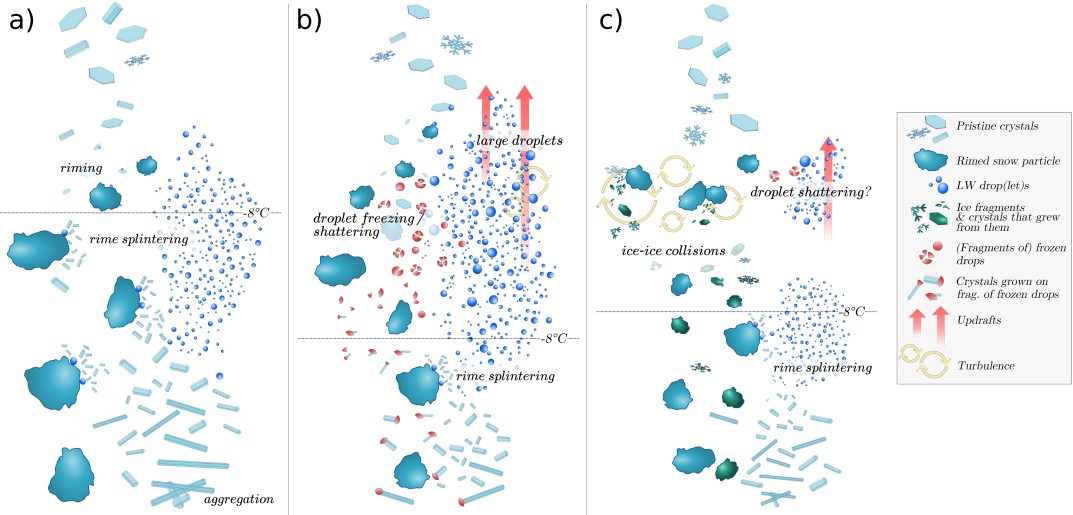

**Figure 11.** Conceptual sketch of the proposed interpretations for the microphysical signatures during the different time frames: (a) 14:50–15:20 UTC, (b) 15:25–15:45 UTC, (c) 16:05–16:30 UTC. Note that HM is also indicated in the lower layers in panels b and c, as it is suspected to occur throughout the event (cf. Sect. 5.1).

size relations, on $Z_e$-IWC relations, and on INP concentrations, which may vary significantly.

All in all, the interpretation of these processes remains hypothetical: an unambiguous demonstration of the occurrence of SIP via a specific process is a challenge that would require more in-situ measurements across scales, to get a full picture of
INP availability and of the interactions between ice (and liquid) particles. Additionally, the information derived from zenith-pointing instruments remains insufficient to grasp the horizontal variability within the precipitating system; it is, for instance, challenging to fully characterize the impact of the orographic terrain in the observations. What remains clear is that different signatures were visible in the remote sensing measurements, calling for distinct interpretations of their possible causes. This also demonstrates the relevance of radar and, in particular, of high-sensitivity Doppler spectral measurements, to investigate
in a detailed way the microphysics of clouds and precipitation. Further studies could include, on the one hand, more involved multi-sensor approaches to confirm the occurrence of SIP, and on the other hand, a generalization of the methods introduced here to gain insight into how frequently such microphysical processes are observed at a given location.

*Data availability.* The data of the ICE GENESIS campaign is available on the *Aeris* platform (https://en.aeris-data.fr/).



## Appendix A: Details on the implementation of LI21

### A1   Diffusional growth model

To model the growth of ice crystals by vapor deposition, we implement the ventilated diffusion growth model presented in Pruppacher and Klett (2010a), following e.g., Hall and Pruppacher (1976), relying on the following equation:

$$\frac{dm}{dt} = \frac{4\pi C \, S_i \, f_v}{(\frac{L_s}{R_v T} - 1)\frac{L_s}{K_{air} T} + \frac{R_v T}{e_{s,ice} \mathcal{D}_v}} \tag{A1}$$

Here and below, all values are given in SI units unless specified otherwise. T is the air temperature, $S_i$ is the supersaturation over ice assuming conditions of saturation with respect to liquid water: $S_i = (e_{s,liq}(T) - e_{s,ice}(T))/e_{s,ice}(T)$, where $e_{s,liq}(T)$ and $e_{s,ice}(T)$ are respectively the saturation vapor pressure over liquid water and over ice (e.g., Huang, 2018). $L_s$ is the latent heat of sublimation (Yau and Rogers, 1989); $K_{air}$ is the thermal conductivity of air, $R_v$ is the gas constant of water vapor and $\mathcal{D}_v = 0.211 \times 10^{-4}(\frac{T}{T_0})^{1.94}\frac{P_0}{P}$ is the molecular diffusion coefficient of water vapor in air (Pruppacher and Klett, 2010a) with $P$ denoting the pressure, $T_0 = 273.15$ K and $P_0 = 1013.25$ hPa.

$f_v$ is the ventilation coefficient, which depends on particle habit: in this study we used the equations of Pruppacher and Klett (2010a) and Ji and Wang (1999) for columnar (CC), plate-like (PLATE) and dendritic (DEN) crystals:

$$f_{v,CC} = 1 - 0.00668\left(\frac{X}{4}\right) + 2.39402\left(\frac{X}{4}\right)^2 + 0.73409\left(\frac{X}{4}\right)^3 - 0.73911\left(\frac{X}{4}\right)^4 \tag{A2}$$

$$f_{v,PLATE} = 1 - 0.6042\left(\frac{X}{10}\right) + 2.79820\left(\frac{X}{10}\right)^2 + 0.31933\left(\frac{X}{10}\right)^3 - 0.06247\left(\frac{X}{10}\right)^4 \tag{A3}$$

$$f_{v,DEN} = 1 + 0.35463\left(\frac{X}{10}\right) + 3.55333\left(\frac{X}{10}\right)^2 \tag{A4}$$

where $X = Sc^{\frac{1}{3}} Re^{\frac{1}{2}}$ depends on the Schmidt number $Sc = 0.632$ and the Reynolds number $Re = \frac{\rho_a L_* v}{\mu_a}$ with $\rho_a$ and $\mu_a$ the density and dynamic viscosity of air. $Re$ in turn relies on a spheroidal model of the ice crystals (prolate for needle-like particles, oblate for planar particles) with $L_*$ the effective aerodynamic size defined as the ratio of the spheroid total surface area to the perimeter of its projection normal to the flow.

The capacitance $C$ is also a function of particle geometry, through the aspect ratio $A_r$ (Pruppacher and Klett, 2010a):

$$C_{obl} = \frac{D\sqrt{1 - A_r^2}}{\arcsin(\sqrt{1 - A_r^2})} \tag{A5}$$

$$C_{prol} = \frac{D}{A_r}\frac{\sqrt{A_r^2 - 1}}{\ln(A_r + \sqrt{A_r^2 - 1})} \tag{A6}$$

$$\tag{A7}$$





We additionally use parameterizations of mass-size and velocity-size relations to propagate Eq. A1 and model the growth of the ice crystals during their fall:

$$v = a_{v,m} \, m^{b_{v,m}} \left( \frac{P_1}{P} \right)^{0.35} \tag{A8}$$

$$v = a_{v,d} \, D^{b_{v,d}} \left( \frac{P_0}{P} \right)^{0.35} \tag{A9}$$

$$m = a_m D^{b_m} \tag{A10}$$

where $P_1 = 8.8 \times 10^4$ Pa, and $a_{v,m}$, $b_{v,m}$, $a_{v,d}$, $b_{v,d}$, $a_m$, $b_m$ are geometry-dependent coefficients listed in Table A1. For columnar crystals: Eq. A8 and the coefficients are from Kajikawa (1976); for planar crystals (plates and dendrites), Eq. A9 and coefficients from Heymsfield and Kajikawa (1987).

| Crystal type | $A_r$ | $a_v$ | $b_v$ | $a_{v,d}$ | $b_{v,d}$ | $a_m$ | $b_m$ |
|---|---|---|---|---|---|---|---|
| COL2 | 2 | 107 | 0.271 | - | - | 0.00929 | 1.8 |
| COL4 | 4 | 162 | 0.302 | - | - | 0.0185 | 1.9 |
| COL8 | 8 | 66 | 0.271 | - | - | 0.00427 | 1.8 |
| DEN | 0.1 | - | - | 5.01 | 0.48 | 0.0232 | 2.29 |
| DEN2 | 0.1 | - | - | 3.29 | 0.11 | 0.242 | 2.53 |
| PLATE | 0.2 | - | - | 29.5 | 0.68 | 1.78 | 2.81 |

**Table A1.** Coefficients of the velocity-size ($v = a_{v,m} \, m^{b_{v,m}}$ or $v = a_{v,d} \, D^{b_{v,d}}$) and mass-size ($m = a_m \, D^{b_m}$) relations, where $m$ is the mass of the crystal, $D$ its maximum dimension and $v$ its terminal velocity. SI units are used.

## A2 Comparison of modeled and estimated terminal velocity

The adequacy of the growth model and microphysical parameterization is verified by comparing the modeled terminal velocity to an estimate of the true one ($v_t$), shown in Fig. A1. In the first implementation of the LI21 method in Sect. 5.1.1, this is done by considering as in Li et al. (2021) that cloud SLW droplets are passive air motion tracers; the settling velocity of the ice particles is then estimated as $v_{t,CC1} = MDV_{CC1} - MDV_{CLW1}$. In the case of Sect. 5.2.2, there is no detected cloud SLW mode that would be fully separated from DL2; to correct for the possible effect of vertical air motion on MDV, we follow Luke et al. (2021) and use the velocity at the edge of the spectrum, corrected with 0.2 m s$^{-1}$ as a rough estimate of typical turbulent spread (the resulting velocity correction is $v_a$). In the last example, the significant air motion and absence of a consistently detected SLW mode make the estimation of $v_t$ much more difficult; Fig. A1 i) illustrates the large difference between MDV$_{DL3}$ and $v_{t,DL3} =$MDV$_{DL3} - v_a$. The estimation of air motion as in Luke et al. (2021) used here to compute $v_a$ is less reliable due to the greater spectral broadening in this turbulent region; as a result, the comparison of modeled vs. estimated $v_t$ cannot be conclusive (Fig. A1 i).

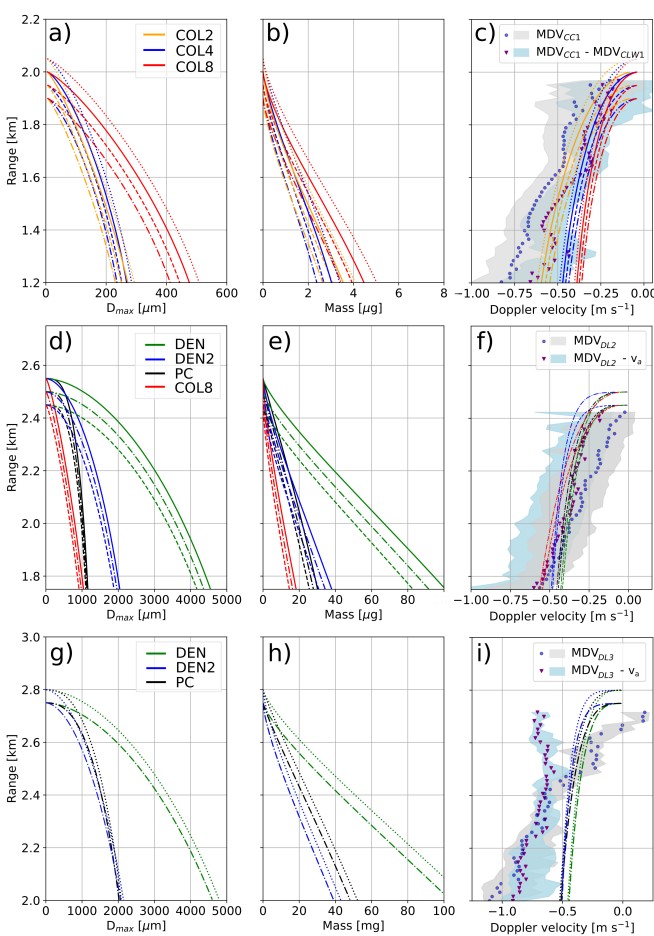

**Figure A1.** Diffusional growth and terminal velocity modeled with the LI21 approach for the various phases (phase 1: a–c, phase 2: d–f, phase 3: g–i). Panels a, d, g: modeled crystal maximum dimension; panels b, e, h: modeled crystal mass (cf. Sect. A1); panels c, f, i: modeled $v_t$, with measured MDV and estimated $v_t$ (cf. Sect. A2, median and interquartile range are shown).



**Appendix B:  LWC content of DL2 - PAMTRA simulations**

In Sect. 5.2.1, signs of the presence of liquid droplets in the mode labeled DL2 were evidenced. This appendix section details
how the radiative transfer model PAMTRA (Mech et al., 2020) was used to simulate the attenuation and reflectivity of a cloud
or drizzle population. A gamma distribution is assumed, with a shape parameter $\mu$ taken in the range -0.5 to 5 (Bringi et al.,
2003)). Simulations are run by varying $\mu$ as well as the LWC and the effective diameter $D_{eff}$, which is the ratio of the third
to the second moment of the particle size distribution. The median volume diameter $D_v$ can be inferred from the effective
diameter as in e.g., Straka (2009); Ulbrich and Atlas (1998). Absorption and scattering coefficients are calculated with Mie
theory, with the liquid water refractive index following Turner et al. (2016). Then, attenuation due to hydrometeors as well as
radar reflectivity at W-band are modeled for a temperature of -10°C. Appendix Figure B1 illustrates the results.

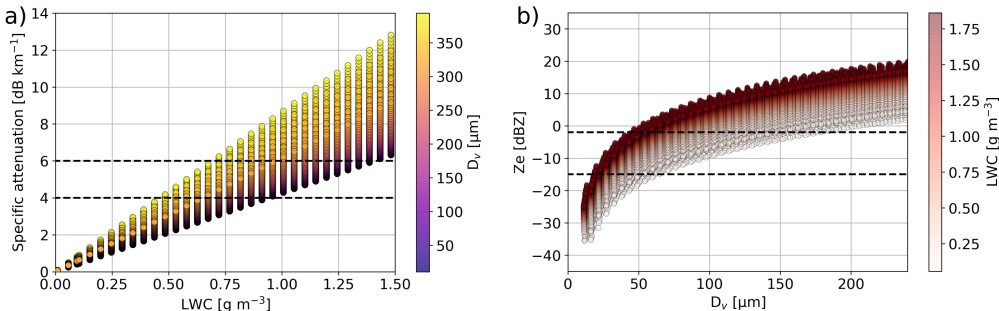

**Figure B1.** PAMTRA simulations of a gamma distribution of liquid droplets with varying parameters (set through $\mu$ = -0.5..5, $D_{eff}$ =
10..300$\mu$m, LWC = 0.01..2 $gm^{-3}$). (a) Specific attenuation due to liquid water vs. LWC, color-coded with $D_v$, (b) $Z_e$ vs. $D_v$, color-coded
with LWC. The black dashed lines indicate the bounds of DL2.



## Appendix C: WRF simulations

Figure C1 illustrates the three nested domains used in the simulations. In Fig. C2, we show the surface variables measured by the automatic weather station at the ground site and the simulated WRF fields.

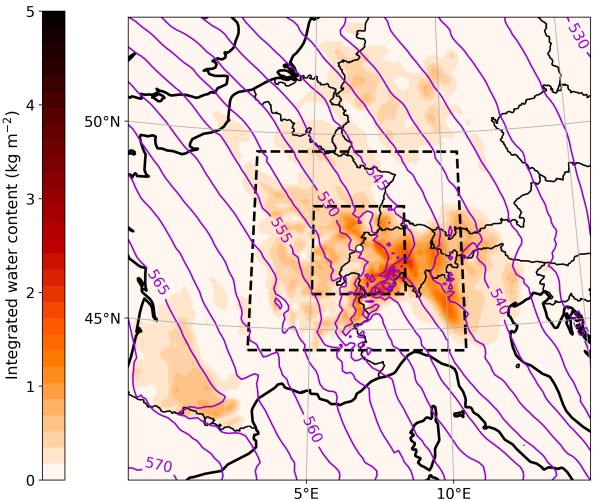

**Figure C1.** WRF simulations of vertically-integrated water content, with geopotential height at hPa (contours; unit: dam) at 12UTC on 27 January. White dot indicates La Chaux-de-Fonds. Dashed boxes show the nested domains.

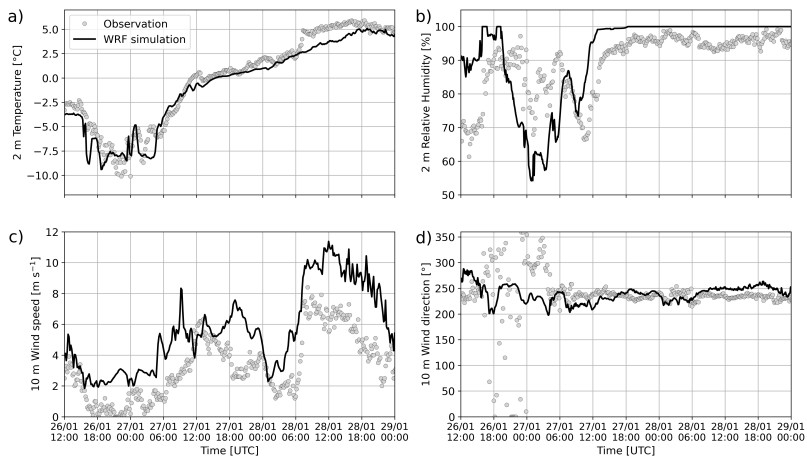

**Figure C2.** WRF simulations and observations (source: MeteoSwiss) of surface meteorological variables: (a) 2-meter temperature; (b) 2-meter relative humidity; (c) 10-meter wind speed; (d) 10-meter wind direction.



*Author contributions.* ACBR and AB conceived the study. PG conducted the WRF simulations with input from AN. LJ, PC and AS processed and analyzed the aircraft observations. ACBR processed and analyzed the radar measurements with input from JG, PG and AB. ACBR wrote the manuscript with contributions from PG, JG and AB. All authors took part in the scientific interpretations and editing of the paper.

*Competing interests.* The authors declare that they have no competing interests.

*Acknowledgements.* This project has received support from the European Union's Horizon 2020 research and innovation program under
grant agreement No 824310 (ICE GENESIS project). PG and AN have received funding from the Horizon 2020 project FORCeS (grant 821205) and the European Research Council project PyroTRACH (grant 726165). Airborne data were obtained using the aircraft managed by SAFIRE, the French facility for airborne research, an infrastructure of the French National Center for Scientific Research (CNRS), Météo-France and the French National Center for Space Studies (CNES). Most of the microphysical in-situ data were collected using instruments from the French Airborne Measurement Platform, a facility partially funded by CNRS/INSU and CNES.



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
