# Peer review of "Distinct secondary ice production processes observed in radar Doppler spectra: insights from a case study"

_EGUsphere, 2023_

## Referee Comment (RC1)

**Review of "Distinct secondary ice production processes observed in radar Doppler spectra: insights from a case study" by Anne-Claire Billault-Roux et al.**

**Overview**

This study is aimed at identifying SIP events based on the analysis of ground-based remote sensing and in-situ observations. The presented measurements were collected from a deep frontal cloud system during the ICE GENESIS campaign. The remote sensing identification of SIP events was performed based on the analysis of primarily W-band radar Doppler spectra and SLDR. In my opinion, identification of SIP mechanisms is hindered by the large spatial separation between the observation point of secondary ice particles and the location of their origin. For this reason, the identification of the SIP mechanisms presented here is not convincing and can even be misleading (major comment #1). However, I see great value in this study in the collected data set combining remote sensing and in-situ measurements and a comprehensive analysis of the radar data. The employed remote sensing technique allows for identifying the presence of cloud particles with different properties (i.e. characteristic shape, Doppler velocity) and identifying SIP events with a high degree of confidence. The observational part of the paper undoubtedly deserves publication. However, the portion related to the identification of the SIP mechanisms is very concerning.

**Recommendation**: I would suggest that the authors to withdraw the paper and rewrite it following the suggestions below. However, if the authors decide to proceed with the existing development, they should address the comments listed below prior to publication.

**Major comments**

1.  The biggest concern in this study is the attempt to identify SIP mechanisms based on the analysis of the remote sensing measurements performed over the ground-based site. The particles which are present in the vertically pointing radar beam at each moment of measurements arrived there from different upstream locations and, therefore, have different ages and have experienced different histories of $RH$ and $T$. The measured secondary ice particles most likely originate many kilometers away from the location of the measurements. Crucially, this means that the conditions required to initiate the specific SIP mechanism that has created these particles most likely will not be persist at the location of their measurement. For example, the small secondary ice particles, which result from ice-ice collisional breakup between graupel will be spatially separated from graupel by a minute, due to the large difference between their fall velocities. Additionally, the secondary ice particles that may be subsequently transported by horizontal wind to the point of remote sensing and measured by the time when the graupel will precipitate down to the ground. Thus, depending on the vertical profile of $RH$ and $T$, for secondary ice particles formed at 2km, it could take more than one hour to precipitate down to the ground. Therefore, the location of production and measurement of secondary ice may be separated by many kilometers depending on the horizontal wind speed. Such large spatial separation and variability of environmental parameters hinders accurate identification of SIP mechanisms and may result in misleading conclusions. This is a serious limitation in identification the of the SIP mechanisms from the vertically pointing ground-based radars. Under these circumstances, it would be relevant to reduce or completely remove sections associated with the identification of SIP mechanisms as highly speculative.

2. **Suggestions:** The authors may consider refocusing the interpretation of the observations on the identification of the conditions of the formation of multimodal Doppler spectra. For example, it can be speculated that the source of secondary ice particles required to produce bimodal Doppler spectra as in Figs.1,5,7 should have a large horizontal and small vertical extensions (pancake type source). It is unlikely that production of secondary ice particles that extended over a large vertical distance (e.g., >1km) would produce patterns like in Figs.1,5,7. The authors may consider replacing the attempts to identify SIP mechanisms by a simulation study to reproduce the radar measurements. I believe you will discover much more interesting things along this line of inquiry. The simulation study will undoubtedly delay the publication. However, in my opinion, this will be worth the effort and increase the scientific value of this study.

3. In-situ measurements obtained in the frame of this study contain plenty of information, which could be used for a scrupulous interpretation and calibration of the radar techniques. Unfortunately, the in-situ data was used in a qualitative way. One of the options, which the authors may consider is to calculate the Doppler spectrum and SLDR based on the analysis of the particle probe measurements and particle image recognition. This would be a solid basis for interpretation of the radar outputs.

4. In Section 5.2.1 the estimated liquid water content ($0.9<LWC<1.4g/m3$) appears to be overly high for the stratiform region in a cold season frontal cloud system. I do not see a reasonable explanation for the formation of such high LWC for this specific case. The problem is aggravated by presence of ice particles, which are expected to rapidly deplete liquid water through the WBF and riming processes. I also suspect that the high value of the measured LWP >800g/m2 may be contributed by the melting layer, which is clearly seen in Fig.4c.

5. Another point of concern is related to instant transition of disk-like particles to columnar ice at approximately 1.9km as shown in Fig.8. An instant transition of particle habits does not sound physically possible. The disk-like and columnar particles are expected to coexist at some range of altitudes like in Fig.7. Could you explore in more detail the retrievals behind this and attempt to explain this case?

6. Recent laboratory studies by Hartmann et al. suggest that the role of the HM-process is overestimated. The authors may consider adding a disclaimer regarding the role of the HM-process. It is worth noting that some past laboratory studies also did not observ SIP during riming (e.g., Hobbs and Burrows, 1966; Aufrermaur and Johnson, 1972).

Aufdermaur, A. N. and Johnson D. A.: Charge separation due to riming in an electric field, Q. J. Roy. Meteor. Soc., 98, 369–382, https://doi.org/10.1002/qj.49709841609, 1972.

Hartmann, S., Seidel, J., Keinert, A., Kiselev, A., Leisner, T., and Stratmann, F.: Secondary ice production - No evidence of a productive rime-splintering mechanisms during dry and wet growth, EGU General Assembly 2023, Vienna, Austria, 24–28 Apr 2023, EGU23-11199, https://doi.org/10.5194/egusphere-egu23-11199 , 2023.

Hobbs, P. V and Burrows, D. A.: The Electrification of an Ice Sphere Moving through Natural Clouds, J. Atmos.Sci., 23, 757–763, https://doi.org/10.1175/1520-0469(1966)023<0757:TEOAIS>2.0.CO;2, 1966.

7. Ground based observation of ice habits and time series of temperature would be a valuable addition to the radar measurements and may facilitate their interpretation.

**Minor comments**
1. Lines 40-41, "*The so-called Hallett-Mossop (HM) rime splintering mechanism (Hallett and Mossop, 1974) occurs as supercooled cloud droplets or drizzle/rain drops rime onto ice particles, generating ice splinters in the process*…" Following the original definition, "*drizzle/rain drops rime*" is not included in the HM process. In fact, riming of drizzle/rain drops on ice is in a gray zone in terms of the type of SIP mechanism it may initiate. Thus, impact of drizzle/rain drops with a much smaller in size ice crystal will result in the initiation of the droplet shattering mechanism rather than HM mechanism.

2. Caption to Figure 1: What is the definition of "fragile aggregates"? I am wondering if the HVPS low resolution imagery is sufficient for identification of mechanical properties of ice particles.

3. Lines 265-268: "*the detection of columnar crystals, at first in a restricted altitude range around 1.5 km…*" and "*a disk-like mode is identified either in restricted altitude ranges…*" Clarify "*restricted altitude range*".

4. Figure 3: What is the dotted line in the vicinity of 4km?

5. Figure 6c: Identification of HVPS images in three green boxes (two top image frames in Fig.6c) is quite ambiguous and not convincing. The resolution and quality of the images are not sufficient to draw such a conclusion. Why don't you use 2DS with a higher pixel image resolution to defend your statement?

6. Figure 6: The circular images in blue boxes do not contain information about the thermodynamic state of these particles. Therefore, their identification as "liquid droplets/drops" in the figure caption is an overstatement.

7. Figure 6: Fix overlayed text in the HVPS titles.

---

## Author Comment (AC2)

**Distinct secondary ice production processes observed in radar Doppler spectra: insights from a case study**

A.-C. Billault-Roux, P. Georgakaki, J. Gehring, L. Jaffeux, P. Coutris, A. Schwarzenboeck, A. Nenes and A. Berne

*egusphere-2023-478*

Responses to reviewers

June 28, 2023

We would like to thank the two anonymous reviewers for their constructive feedback and useful suggestions. These helped improve the manuscript by clarifying some of the underlying assumptions and discussing more thoroughly certain observations.

In addition to the changes made to address the comments of the reviewers, a few minor modifications of the text were made, and most figures were redone using colorblind-friendly color schemes.

In the following, each section corresponds to the comments of a referee.

| *italic text* | *Comments of the reviewers* |
|---|---|
| roman text | Our answers or comments |
| blue text | Verbatim quotes from the revised manuscript |

**1   Reviewer #1**

*This study is aimed at identifying SIP events based on the analysis of ground-based remote sensing and in-situ observations. The presented measurements were collected from a deep frontal cloud system during the ICE GENESIS campaign. The remote sensing identification of SIP events was performed based on the analysis of primarily W-band radar Doppler spectra and SLDR. In my opinion, identification of SIP mechanisms is hindered by the large spatial separation between the observation point of secondary ice particles and the location of their origin. For this reason, the identification of the SIP mechanisms presented here is not convincing and can even be misleading (major comment #1). However, I see great value in this study in the collected data set combining remote sensing and in-situ measurements and a comprehensive analysis of the radar data. The employed remote sensing technique allows for identifying the presence of cloud particles with different properties (i.e. characteristic shape, Doppler velocity) and identifying SIP events with a high degree of confidence. The observational part of the paper undoubtedly deserves publication. However, the portion related to the identification of the SIP mechanisms is very concerning.*

*Recommendation: I would suggest that the authors to withdraw the paper and rewrite it following the suggestions below. However, if the authors decide to proceed with the existing development, they should address the comments listed below prior to publication.*

We thank the reviewer for their assessment of the manuscript and for highlighting the interest of the observations during this case study. We also thank them for raising this important point which was

not discussed in-depth in the original version of the manuscript, i.e., the question of how the advection of the precipitating system may impact the interpretation of the radar observations in terms of microphysical processes. We fully agree that some caution is needed on this point, and we included a subsection to discuss this. Nonetheless, we believe that at least for this snowfall event, a few strong arguments can be made in favor of such interpretations, as we detail in our response to major comment #1.
* * *
**Major comments**

1. *The biggest concern in this study is the attempt to identify SIP mechanisms based on the analysis of the remote sensing measurements performed over the ground-based site. The particles which are present in the vertically pointing radar beam at each moment of measurements arrived there from different upstream locations and, therefore, have different ages and have experienced different histories of RH and T. The measured secondary ice particles most likely originate many kilometers away from the location of the measurements. Crucially, this means that the conditions required to initiate the specific SIP mechanism that has created these particles most likely will not be persist at the location of their measurement. For example, the small secondary ice particles, which result from ice-ice collisional breakup between graupel will be spatially separated from graupel by a minute, due to the large difference between their fall velocities. Additionally, the secondary ice particles that may be subsequently transported by horizontal wind to the point of remote sensing and measured by the time when the graupel will precipitate down to the ground. Thus, depending on the vertical profile of RH and T, for secondary ice particles formed at 2km, it could take more than one hour to precipitate down to the ground. Therefore, the location of production and measurement of secondary ice may be separated by many kilometers depending on the horizontal wind speed. Such large spatial separation and variability of environmental parameters hinders accurate identification of SIP mechanisms and may result in misleading conclusions. This is a serious limitation in identification the of the SIP mechanisms from the vertically pointing ground-based radars. Under these circumstances, it would be relevant to reduce or completely remove sections associated with the identification of SIP mechanisms as highly speculative.*

We thank the reviewer for pointing out this limitation, which is common in radar-based studies. We agree with the reviewer that more arguments are needed to justify the interpretation of microphysical processes from measurements of vertically-pointing radar, because of the advection component. However, we believe that several arguments can be raised to show that, in this case at least, it is reasonable to propose interpretations of the radar signatures in terms of microphysical processes — also in view of the existing literature on the topic, as discussed further.

The first argument is related to the **temporal homogeneity** of the signatures and the **absence of significant directional wind shear** in the altitude range which is the focus of our analysis (1 to 4 km above ground level). When no significant directional shear is present, an analysis of processes along *fall streaks* can be conducted as proposed by Kalesse et al. (2016); Pfitzenmaier et al. (2017, 2018). Pfitzenmaier et al. (2017) defines a fall streak as "the path of a particle population obtained from the observation of its own motion"; Kalesse et al. (2016) and Pfitzenmaier et al. (2018) rely on fall streak corrected spectrograms to "investigate the changes in the spectra due to microphysical changes of the tracked ice particle distribution" (Pfitzenmaier et al., 2018). The fall streak reconstruction allows to take into account the effect of the advection of the cloud/precipitation system in the absence of directional wind shear. This is a good approximation in our case study in the levels above $\sim 1$ km above ground, where the wind direction is approximately constant in the region of La Chaux-de-Fonds (LCDF) throughout the event (Fig. R1 below, now included in the Appendices of the revised manuscript). We therefore implement the fall streak tracking using wind speed profiles from WRF and with Eq. 1 from Kalesse et al. (2016). This algorithm is run using the mean Doppler velocity of (i) the rimer mode (background ice,

fast-falling population) and (ii) when present, the secondary mode of the Doppler spectra. While this fall streak tracking still differs from the Lagrangian trajectory of a population of particles, it provides some robust insights into the spatio-temporal trajectory of a particle population. In our case, this analysis highlights two things. First, it shows that the time extent of fall streaks associated with the new ice populations are within the time frames of the phases we analyze. In other words, the phases we focus on are long enough that we can reasonably consider that the effect of advection does not contaminate the interpretation of the processes within these phases. Secondly, if we focus on the spectrograms reconstructed along those fall streaks, similar observations come through (compared to looking at vertical profiles from single-timestep spectrograms) as illustrated by Fig. R2 below (now included in the Appendix of the manuscript). This shows that the temporal homogeneity is sufficient to allow for the analysis of single-time-step spectrograms. Additionally, we recall that most of our analyses are based on the *statistics* of the radar moments (median + interquartile range) within each of these phases: this ensures that the interpretations are not contaminated by small-scale advection-related inhomogeneities.

[Figure]

Figure R1: Horizontal wind direction from WRF simulations above LCDF

[Figure]

Figure R2: (a) $Z_{e,W}$ timeseries with retrieved fall streaks of faster-falling and secondary modes corresponding to different particle types. (b)–(d) Doppler spectrograms reconstructed along the corresponding fall streaks, with sZe and SLDR.

The second argument—even more explicitly related to the comment of the reviewer—comes from the **spatial homogeneity** of the precipitating system at the scales we are considering. Going back to the example proposed by the reviewer of particles precipitating for several hours, and hence over large spatial distances: we would like to highlight that the altitude ranges which our

analysis focuses on are narrower than that mentioned by the reviewer (2 km). During each phase, we focus on the altitude range where the new ice populations (new Doppler modes) are formed; these regions extend over no more than $\sim 500$ m. An order of magnitude calculation shows that such particles may take around 15-20 min to precipitate with $0.5$ m s$^{-1}$ fall speed; with the mean horizontal wind of 20 m s$^{-1}$, this corresponds to a horizontal advection of $\sim$20 km. Focusing now on modeled fields with the WRF cross-sections as in Fig. 9 (see also Fig. R3), it comes across that there is little variability of the main atmospheric variables within this spatial scale, at least around the considered altitudes ($>$1 km above LCDF, i.e. $>$2 km ASL) and windward of LCDF [further downwind of LCDF, a hydraulic jump in the lee of the Jura mountains creates a significant heterogeneity]. The distinct cloud layers, shear and humidity profiles are largely homogeneous, with only mild terrain-related fluctuations. The terrain windward of the Jura is characterized with a gentle upslope (which is accurately represented in the model) with little heterogeneity, hence no abrupt change of atmospheric conditions is expected, nor modeled, within this narrow 20-km region. As a result, the conditions of formation of particles having precipitated over a few hundreds of meters when reaching LCDF are comparable to the conditions of formation above LCDF. This legitimates the rationale that we follow when proposing interpretations of SIP processes.

[Figure]

Figure R3: Examples of cross-sections from WRF simulations (following mean wind direction, 315°). (a) and (c): Temperature (color shade) with saturation w.r.t. ice (grey hatched zone); (b) and (d): Relative humidity w.r.t. liquid water (color shade) with 99% saturation w.r.t. liquid water (white dashed contour). All panels include isentrope contours and wind barbs.

More generally, we would like to mention that multiple studies in the past have relied on Doppler spectra from zenith-pointing radars to investigate ice production and growth mechanisms, including SIP processes (e.g., to list a few Zawadzki et al., 2001; Rambukkange et al., 2011; Verlinde et al., 2013; Kalesse et al., 2016; Pfitzenmaier et al., 2018; Oue et al., 2018; Li et al., 2021; Ramelli et al., 2021; Luke et al., 2021). Our study therefore builds on a solid literature basis in terms of studying precipitation microphysics with zenith-pointing radar. Naturally, we agree with the reviewer that radar-based studies have their limitations and that they cannot provide a fully unequivocal demonstration that a given SIP process is taking place. For this reason, we have tried in multiple places of the manuscript to underline these limitations and to state that an unambiguous identification of a SIP mechanism with radar measurements is difficult. Nonetheless, we believe that what makes our inferences reasonable is that, for each of the identified phases, we rely on several independent measurements / sources of information (e.g., Doppler spectra, dual-frequency reflectivity ratio (DFR), liquid water path (LWP), modeled atmospheric fields, aircraft images) and show that they all concur with the proposed interpretations.

As a last remark to respond to this comment, we would like to highlight that one original aspect of our study is to identify possible SIP signatures in different temperature ranges: the first phase is within the columnar growth region, but the new ice produced in phases II and III is formed at higher altitudes and at colder temperatures and has non-columnar shapes. This could possibly be said even without more subtle discussions on the advection issue. With the current state of knowledge on SIP (and especially of its identification through radar measurements), this is

already a valuable information: the processes in phases II and III are very unlikely to be related to Hallett-Mossop. Then, the spatio-temporal behavior of the new ice particles is strikingly different in phases II and III, which naturally calls to formulate hypotheses on what may be happening: this is the path that we tried to follow.

We now include a subsection (Sect. 5.1) to discuss these various aspects. This is complemented with a new Appendix section to detail the fall streak reconstruction.

[Section 5.1 – Spatial and temporal homogeneity of the cloud system] Before delving into the analysis of microphysical signatures in these three phases, a few words should be added regarding necessary caution in the interpretation of vertically-pointing radar measurements. In general, a single Doppler spectrogram at a given time step should not be interpreted as the microphysical history of particles from cloud top to ground: because of the advection of the cloud system, particles observed close to the ground were formed windward and were advected toward the observation site as they precipitated. To avoid any misleading intepretations, we verify two key aspects of this issue.

The first aspect is related to the temporal homogeneity of the radar measurements and the absence of significant directional wind shear in the altitude range which is the focus of our analysis (1 to 4 km, see Appendix A, Fig. A1b). When no strong directional shear is present, an analysis of microphysical processes can be performed along fall streaks, which reveal the spatio-temporal path of a particle population (e.g., Kalesse et al., 2016; Pfitzenmaier et al., 2017, 2018); Doppler spectra can then be remapped along these fall streak paths. Reconstructed fall streaks within each phase of the event are shown in Appendix A, from which two main conclusions are drawn. On the one hand, the temporal extent of the fall streaks is well within the time frame of each phase: this implies that the phases we consider are long enough for statistics of radar variables to be representative of microphysical processes within these time periods. On the other hand, the Doppler spectrograms that are reconstructed along the slanted fall streaks, although noisier than the original ones, yield similar interpretations in terms of the coexistence of various particle populations. This highlights the temporal homogeneity of the system and legitimates the analysis of single-time step spectrograms as done in further sections.

The second aspect is related to the windward horizontal spatial homogeneity of the cloud system. Considering an example that is representative of this case study, with particles precipitating over $\sim 500$ m, with a fall speed of $\sim 0.5$ m s$^{-1}$, and advected by a horizontal wind of $\sim 20$ m s$^{-1}$; the ice particles would have traveled a horizontal distance of $\sim 20$ km from the location of their formation to LCDF. The modeled WRF fields reveal that in the windward direction of LCDF, and at the altitudes of interest (above $\sim 2$ km asl), there is only a mild variability of the main atmospheric variables within this spatial scale: the humidity and temperature profiles during the formation and growth of the particles windward are similar to the ones over the radar site. This can be seen for instance in Fig. 9b–d further on. In this context, it is thus legitimate to investigate the microphysical processes behind the signatures observed in the vertically-pointing radar measurements. It should be underlined that such conditions may not always be satisfied, and this may challenge the interpretation of the radar fields in more complex atmospheric settings.

[Appendix A1 - Details on the fall streak tracking] In order to assess the validity of the approach, i.e., investigate microphysical processes based on radar signatures during 3 phases of the event, a fall streak tracking algorithm was implemented as explained in Sect. 5.1. The method introduced by Kalesse et al. (2016) was used, following their Eq. 1. The horizontal wind speed is taken from WRF simulations (Fig. A1a); the method can be implemented as no significant wind shear is present in the altitude range of interest i.e., 1–4 km (Fig. A1b). The Doppler velocity is taken to be either the one of the fast-falling mode (dashed fall streaks in Fig. A2a) or of the secondary mode (full lines in Fig. A2a), in order to follow the spatio-temporal trajectory of either population. In Fig. A2b–d, examples of Doppler spectrograms reconstructed along slanted fall streaks (of secondary modes) are shown, for each of the three phases. In each case, although

the spectrograms are noisier, the same features are observed as in the spectrograms shown in Figs. 5,7,10.

2. *Suggestions: The authors may consider refocusing the interpretation of the observations on the identification of the conditions of the formation of multimodal Doppler spectra. For example, it can be speculated that the source of secondary ice particles required to produce bimodal Doppler spectra as in Figs.1,5,7 should have a large horizontal and small vertical extensions (pancake type source). It is unlikely that production of secondary ice particles that extended over a large vertical distance (e.g., >1km) would produce patterns like in Figs.1,5,7. The authors may consider replacing the attempts to identify SIP mechanisms by a simulation study to reproduce the radar measurements. I believe you will discover much more interesting things along this line of inquiry. The simulation study will undoubtedly delay the publication. However, in my opinion, this will be worth the effort and increase the scientific value of this study.*

Conducting a modeling study of this snowfall event to try and reproduce the radar measurements would undeniably be an interesting approach. In view of the arguments that we put forward in our response to the reviewer's first comment, we believe that our observational approach is also a valid one. We would like to highlight some of the challenges related to a modeling perspective on such a complex case study.

One challenge comes from accurately describing the SIP processes, whose efficiency is debated: this is the case for all the dominant processes, as highlighted in Korolev and Leisner (2020) (see e.g., their Tables 1 and 2 and Sect. 4). Adequately modeling these processes thus comes with non-negligible uncertainties, and there is to this day a recognized need for more observational and experimental studies to improve the quantitative description of SIP in models (e.g., Morrison et al., 2020). Additionally, some of the signatures we observe are highly localized, such as the generating cells of phase III, and are very difficult to reproduce within a modeling study.

Another major challenge comes from the forward modeling of radar measurements. While reasonable results may be obtained in the forward modeling of radar reflectivity, the full dual-polarization Doppler spectrum—which is the crux of our dataset—cannot to our knowledge be simulated with tools available in the literature (e.g., Oue et al., 2020; Mech et al., 2020; Ori et al., 2021). Difficulties arise from the non-Rayleigh scattering behavior of snow particles at W-band, for which a spheroidal approximation is insufficient (Leinonen et al., 2012). Additionally, a bulk microphysics scheme would also be insufficient to reproduce the observed multimodal behavior.

Because of these challenges, and because of the propagation of uncertainties that they bring, we believe that this is not necessarily a more robust approach than our observation-based interpretations. Naturally, we fully agree that a modeling approach that addresses these difficulties would be of high interest; to open up the discussion to these perspectives, and also following a comment of Reviewer #2, we added a few sentences to the conclusions:

As highlighted by e.g., Sinclair et al. (2016); Young et al. (2019), such case studies where SIP is presumed to be active may also serve to evaluate and improve the microphysical parameterization of SIP processes within numerical weather models. Along this line, further work may include more modeling-oriented approaches, including the forward modeling of radar fields, although this in turn comes with non-trivial questions regarding e.g., the representation of the scattering properties and terminal velocities of the different particle types.

3. *In-situ measurements obtained in the frame of this study contain plenty of information, which could be used for a scrupulous interpretation and calibration of the radar techniques. Unfortunately, the in-situ data was used in a qualitative way. One of the options, which the authors may*

*consider is to calculate the Doppler spectrum and SLDR based on the analysis of the particle probe measurements and particle image recognition. This would be a solid basis for interpretation of the radar outputs.*

We thank the reviewer for this comment. Indeed, as we worked on this study, we carefully considered whether the aircraft measurements could be used in a more quantitative way in combination with the radar observations.

In fact, several issues appear with such an approach for this case study. The first one comes from the flight strategy itself (chosen for other reasons, see Billault-Roux et al., 2023). The aircraft flight levels were mostly lower than the altitude of the formation of the ice particles that we observe (target temperature range of the aircraft: -10 to +2°C). Thus, the aircraft does not inform us exactly on the zone where the new ice is being formed but captures particles which have undergone additional growth processes, resulting in changes of the particle size distribution (PSD). Then, there is some horizontal variability in the small-scale precipitation structures: because the aircraft flies in ∼25 km constant-altitude legs (in the SW-NE direction), its measurements cannot be compared to the radar data, except at the few time steps where the aircraft overpasses the ground site. During the phase of the event which we focus on, only 6 valid points of comparison are available (not necessarily at the altitudes of interest): thus, the quantitative information that can be extracted for a radar / in-situ comparison is very limited.

Another main issue is related to the simulation of radar fields with this high precision (dual-polarization and spectral), see response to Comment #2. In addition to difficulties in the representation of scattering properties, the particle habit identification is not available for all particles observed by the aircraft but only for the larger sizes (2D-S: >200 $\mu$m, PIP: >2 mm). Overall, this means that simulating radar Doppler spectra from aircraft PSDs is highly uncertain.

Regarding the last part of the reviewer's comment, we would like to underline that the reader can have some confidence in the interpretation of particle types that we propose. It is well established in literature that at zenith angle, columnar crystals are the only particle type that produces high (S)LDR values (outside of the melting layer), while other pristine crystals (e.g. plates) and liquid water drops have a very low depolarization (e.g., Matrosov, 1991; Aydin and Tang, 1997; Matrosov et al., 2001; Reinking et al., 2002; Matrosov et al., 2012; Oue et al., 2015; Myagkov et al., 2016; Ryzhkov and Zrnic, 2019; Li and Moisseev, 2020; Li et al., 2021); this is the primary information that we use. Similarly, using a reflectivity threshold combined with a condition on mean Doppler velocity to identify cloud liquid water is commonly done (reinforced with a condition on (S)LDR, when available) (e.g., Frisch et al., 1995; Shupe et al., 2004; Kogan et al., 2005; Kalesse et al., 2016; Li and Moisseev, 2019; Li et al., 2021; von Terzi et al., 2022). After this step, the analysis of aircraft in-situ images in Fig. 1 (where we also incorporate quantitative information from the PIP classification) gives all the more confidence in the proposed identification of particle habits from the polarimetric Doppler spectra.

To clarify to the reader why we do not rely on the aircraft data in a more quantitative way, we added a sentence to Sect. 2.2:

In this study, the aircraft observations are chiefly used as a complementary source of information to analyze particle habits and the possible occurrence of microphysical processes. Because only a few points are available when the aircraft overpasses the radar, the possibilities for a joint quantitative analysis of radar and aircraft measurements were limited.
* * *
4. *In Section 5.2.1 the estimated liquid water content (0.9<LWC<1.4g/m3) appears to be overly high for the stratiform region in a cold season frontal cloud system. I do not see a reasonable explanation for the formation of such high LWC for this specific case. The problem is aggravated*

*by presence of ice particles, which are expected to rapidly deplete liquid water through the WBF and riming processes. I also suspect that the high value of the measured LWP >800g/m2 may be contributed by the melting layer, which is clearly seen in Fig.4c.*

We are grateful to the reviewer for having pointed out these high LWC values. Indeed, this is related to a mistake on our side, due to a misinterpretation of the PAMTRA fields; what we used was the one-way attenuation instead of the two-way. Hence, the correct values obtained with the proposed method are twice smaller, i.e., 0.45 to 0.7 g m$^{-3}$. This marginally affects the retrieved $D_v$ (40 to 90 $\mu$m instead of 35 to 70 $\mu$m). Note that these LWC values are consistent with similar literature approaches as detailed for instance in Tridon et al. (2020): there, through different methods the authors find a two-way PIA of 4 dB to be equivalent to LWP of 600 g m$^{-2}$. This is also compatible with LWC values measured during the flight (up to 0.4 g m$^{-3}$), although no direct comparison can be made as no overpass took place during phase II in the altitude range where the strong W-band attenuation is seen. Lastly, these LWC values are in agreement with the modeled WRF fields which exceed 0.5 g m$^{-3}$ in this height range and around this time (Fig. 9a). Overall, this modification does not alter the reasoning that was conducted: the LWC values derived from the change in DFR are still high and call for some discussions regarding the mechanisms by which the LW is sustained: this is the topic of Sect. 5.2.3 (new Sect. 5.3.3).

We then rely on the measurements of DL2 as constraints on reflectivity (between -15 and -2 dBZ), specific attenuation (between 4 and 6 dB km$^{-1}$, calculated from the increase of DFR within the DL2 layer; see Fig. 7c) [...] this translates into bounds on LWC and median volume diameter ($D_v$, such that half of the volume of water is contained in droplets smaller than $D_v$): 0.45 g m$^{-3}$ < LWC < 0.7 g m$^{-3}$, and 40 $\mu$m < $D_v$ < 90 $\mu$m.

Besides, we fully agree with the reviewer that the LWP is affected by the partial melting layer (ML) lower down; this is precisely the reason why we decided to rely mostly on the analysis of the DFR and of how this DFR varies in the upper part of the cloud (reflecting differential attenuation in this region), **within** the DL2 layer. To clarify this, we included a statement regarding the effect of the partial ML on LWP; we also clarify that the values are computed using the change of DFR within the altitudes of interest (so not affected by LW lower down). We still think that mentioning the very high LWP values is relevant, as it gives a complementary support to our interpretations: if the LWP were low during this time frame, this would strongly contradict our inferences.

Secondly, the LWP time series (Fig. 4e) reaches remarkably large (> 800 g m$^{-2}$) values during this time frame. The LWP retrieval does not inform on the altitude of the liquid cloud layers—here, it is likely also affected by the partial melting layer around 500 m—but it does confirm a significant presence of liquid water in the atmospheric column during this period.
* * *
5. *Another point of concern is related to instant transition of disk-like particles to columnar ice at approximately 1.9 km as shown in Fig.8. An instant transition of particle habits does not sound physically possible. The disk-like and columnar particles are expected to coexist at some range of altitudes like in Fig.7. Could you explore in more detail the retrievals behind this and attempt to explain this case?*

We thank the reviewer for raising this point. As a first side remark, we would like to recall an important difference between Fig. 7 and Fig. 8: the former illustrates the statistics (median + IQR) of the Doppler peak moments during phase 2; while the latter (Fig. 8) focuses on a single time step (spectrogram in Fig. 8d and e). Then, the observation that the secondary mode is identified as "disk-like" at higher levels and as "columnar" in lower levels comes from the fact

that columnar crystals are being produced at lower altitudes. [NB This likely corresponds to the mechanism described in Sect. 5.1 (new Sect. 5.2), where we also state that the process seems to occur throughout the event, as columnar crystals are persistently identified below ∼-8°C. The new columnar crystals may also be growing onto small ice fragments sedimenting from above.] In any case, the radar signature of these new columnar crystals is entangled with the signature of the "DL" crystals precipitating from above, as their fall velocities are not well separated: this results in a single Doppler mode containing diverse shapes of crystals. At some point, the columnar crystals (strongly depolarizing) become dominant in the polarimetric signature. This corresponds to the transition in the labeling of the peaks. Some understanding of this transition can be gained from the SLDR spectrogram (Fig. 8e) where high SLDR values are first visible only on the right edge of the mode, then through most of it, as columns grow to larger sizes.

We agree with the reviewer that this feature deserves an explanation. We therefore include the following changes:

[Sect. 3.2] We underline that this classification method only allows to label the particle type which is dominant in the radar signature: in some cases, distinct particle habits may coexist that do not result in different Doppler modes because of their similar fall velocities or because of turbulent broadening; the labeling routine will then be sensitive to the dominant particle type (e.g., cloud droplets may not be identified even if they are present).

[New Sect. 5.3.2] Note that the abrupt change of the classification output around 1.8 km from disk-like to columnar does not mean that the particles themselves transition from one type to the other; rather, this is due to disk-like particles (precipitating from above) and newly-formed columnar crystals being entangled in a single Doppler mode. Around 1.8 km, the columnar crystals start becoming dominant in the radar signal because of their strong depolarization, which results in the change of label for the secondary mode.
* * *
6. *Recent laboratory studies by Hartmann et al. suggest that the role of the HM-process is overestimated. The authors may consider adding a disclaimer regarding the role of the HM-process. It is worth noting that some past laboratory studies also did not observe SIP during riming (e.g., Hobbs and Burrows, 1966; Aufrermaur and Johnson, 1972).*

We thank the reviewer for providing this new reference. In the introduction, we nuance the efficiency of the HM process and highlight the ongoing debate:

HM is active between -8°C and -3°C, with a maximum efficiency around -5°C. Note that the efficiency of this process is still questioned due to contrasted experimental results (e.g., Korolev and Leisner, 2020 and references therein; Hartmann et al., 2023).
* * *
7. *Ground based observation of ice habits and time series of temperature would be a valuable addition to the radar measurements and may facilitate their interpretation.*

Thank you for this suggestion. Regarding the first aspect: ground-based observations of ice habits are indeed an interesting complementary information that can help understand the microphysical history of particles. In this specific case study, one major obstacle comes from the fact that air temperature at ground level is slightly above 0°C during the time frame of interest. This results in particles which are starting to melt and whose original habit is more difficult to ascertain from the available ground-based instrument (multi-angle snowflake camera, MASC, Garrett

et al., 2012, see e.g., Fig. R4a, which shows a melting aggregate whose original composition is hard to assess). Nonetheless, a few clear images of heavily rimed particles and of aggregates of columns were recorded which support the radar-based inferences on particle types (Fig. R4b–d).

[Figure]

Figure R4: Examples of MASC images with manually-added labels. The red bar corresponds to 1 mm.

Another, more general, obstacle to the joint interpretation of ground-based observations and vertically pointing measurements comes from the limitation stated by the reviewer in their major comment #1, related to the advection of the precipitating system. Unlike the radar-based analysis which can be conducted in a small altitude span, comparing measurements aloft to observations at the ground implies a much larger spatial separation. We thus preferred to focus on the particle morphologies observed by aircraft probes which (1) are not affected by the warm ground-level temperature and (2) are closer in altitude to the region of interest so less prone to possible advection-related "mismatches". We did not include the MASC observations in the manuscript as they are most of the time difficult to read and, when they are interpretable, they provide conclusions similar to the ones obtained from aircraft OAP images, without bringing any real additional information in our opinion.

Regarding the second aspect: we agree with the reviewer that information on ground temperature is a relevant addition, and we therefore complemented Fig. 3 with a time series of 2-m temperature. Note that the temperature time series for the entire event (beyond the part that is the focus of this study) as well as other ground-based measurements of meteorological variables are also in Appendix Fig. C2 (D2 of the revised manuscript).

[Added panel d of Fig. 3, with corresponding caption.] (d) 2-m temperature from MeteoSwiss weather station (located 500 m away from the radar site).

**Minor comments**

1. *Lines 40-41, "The so-called Hallett-Mossop (HM) rime splintering mechanism (Hallett and Mossop, 1974) occurs as supercooled cloud droplets or drizzle/rain drops rime onto ice particles, generating ice splinters in the process..." Following the original definition, "drizzle/rain drops rime" is not included in the HM process. In fact, riming of drizzle/rain drops on ice is in a gray zone in terms of the type of SIP mechanism it may initiate. Thus, impact of drizzle/rain drops with a much smaller in size ice crystal will result in the initiation of the droplet shattering mechanism rather than HM mechanism.*

   We thank the reviewer for highlighting this. We initially formulated the definition this way because of the unclear boundary between cloud droplets and drizzle/ rain drops: for instance, the latter are typically described with gamma PSDs, and therefore also potentially cover a size range corresponding to small liquid drops that could be involved in HM. To avoid any confusion, we rephrased:

   The so-called Hallett-Mossop (HM) rime splintering mechanism (Hallett and Mossop, 1974) occurs as SLW droplets rime onto ice particles, generating ice splinters in the process (...).
* * *
2. *Caption to Figure 1: What is the definition of "fragile aggregates"? I am wondering if the HVPS low resolution imagery is sufficient for identification of mechanical properties of ice particles.*

   This wording is the one introduced in Jaffeux et al. (2022), and denotes weakly-bounded crystals. Note that this classification is not based on HVPS but on PIP images, with a higher resolution. We adapted the caption to re-state the definition of fragile aggregates there.

   [In the text] The independent PIP-based morphological classification (Jaffeux et al., 2022, for particles with a maximum dimension greater than 2 mm,) (...) aggregates which are either distinctly classified as made of columns and needles, or simply labeled as fragile, which denote weakly bounded crystals).

   [Caption] Habit classification from PIP images at the same time step (16:15 UTC). HP: hexagonal planar crystals; GR: graupel; RA: rimed aggregates, FA: fragile aggregates **(weakly bounded)**, CA: aggregates of columns/needles, CC: columnar crystals (columns/needles).
* * *
3. *Lines 265-268: "the detection of columnar crystals, at first in a restricted altitude range around 1.5 km..." and "a disk-like mode is identified either in restricted altitude ranges..." Clarify "restricted altitude range"*

   Thank you for noting this unclear phrase; we now include more precise statements:

   [...] columnar crystals, at first in a restricted altitude range around 1.5 km (+/- 250 m, ~13:15 to 15:00 UTC), [...]

   [...] a disk-like mode is identified either in restricted altitude ranges (e.g., 15:00 UTC, ~2 km +/- 300 m) [...]

4. *Figure 3: What is the dotted line in the vicinity of 4km?*

   Thank you for pointing this out, the label of this line went missing in the figure display; it corresponds to the -20°C isotherm. We double-checked that all the isotherms are correctly labeled in the new version of the figure.

   [New version of Fig. 3.]

5. *Figure 6c: Identification of HVPS images in three green boxes (two top image frames in Fig.6c) is quite ambiguous and not convincing. The resolution and quality of the images are not sufficient to draw such a conclusion. Why don't you use 2DS with a higher pixel image resolution to defend your statement?*

   The 2D-S unfortunately has a rather small maximum sample size (1.28 mm), which is smaller than the size of most grown dendritic crystals: this is why no such crystal was visible in the 2D-S images of Fig. 6c of the original manuscript. In the revised version, we replace the 2D-S images with PIP images (maximum size: 6.4 mm) in the lower part of Fig. 6c. Note that with this modification, panel c cannot be directly compared to panels a and b where we kept the 2D-S images, which are more appropriate. With this modification, we can also identify in the PIP data certain crystals which have a morphology close to dendrites or fragments of dendrites.

   [New Fig. 6 and corresponding caption]. Aircraft OAP images for the three time frames: (a) 15:05 / 1400 m (HVPS and 2D-S); (b) 15:40 / 1150 m (HVPS and 2D-S); and (c) 16:20 / 1720 m (HVPS and PIP) [...]. The vertical bar in the PIP images (lower part of panel c) corresponds to 6.4 mm. In panel c, PIP images are included instead of 2D-S as the size range of the latter is too small to capture grown dendritic crystals.

6. *Figure 6: The circular images in blue boxes do not contain information about the thermodynamic state of these particles. Therefore, their identification as "liquid droplets/drops" in the figure caption is an overstatement.*

   The identification of particle type from OAP images is indeed not a straightforward matter. Several studies have been dedicated to modeling the diffraction patterns of different particle types, including spherical liquid droplets or ice crystals, supported with both theoretical and experimental work (e.g., Heymsfield and Baumgardner, 1985; Joe and List, 1987; Korolev, 2007; Vaillant De Guélis et al., 2019, to cite a few). Based on these findings, it is established that the diffraction pattern of spherical drops is a dark disk with a central white spot (while shapes of ice particles result in more complex figures); this is the criterion that we used when labeling the droplets in Fig. 6. We now explain this in the text (see below); we also made a small change to Fig. 6 as one of the identified droplets did not correspond exactly to the description.

   These are identified through their known diffraction pattern, resulting in a dark disk with a central white spot (e.g., Korolev, 2007)

7. *Figure 6: Fix overlayed text in the HVPS titles.*

   Thank you for pointing out this issue in the display. This was fixed in the new version of Fig. 6.

**2 Reviewer #2**

*The article presents a case study of a precipitation event that took place during ICE GENESIS campaign. The main focus of the study is to investigate whether SIP took place during this event and what potential SIP processes were responsible for observed signatures. The authors have analyzed W-band Cloud radar Doppler spectra for this purpose.*

*General comments: In the recent years there have a number of studies where polarimetric Doppler radar spectra were used to identify formation of ice particles and linking it to SIP. In our community it is, unfortunately, common that any spectral bimodality is immediately denoted as SIP. I would like to thank the authors for not following this path and using two distinct methods that can be used to identify potential SIP production. Because the polarimetric signature of columnar-ice production is easy to identify, many studies have mainly focused on that region that coincides with the temperature region where H-M rime splintering processes is expected to take place. That is why I was especially was interested to read the section 5.2.2 where the case of formation of "disk-like" crystals was discussed. Overall, the findings presented in the article are consistent with the previous studies.*

*I find it a missed opportunity that the WRF modeling was only used to provide temperature and wind information. It would be interested to see whether H-M parametrization in WRF was able to identify and accurately represent the observations (see Sinclair et al. (2016) as an example). As there are concerns of the validity of H-M, this study would have been interesting. Hopefully, the authors would have an opportunity to perform such an analysis in the future.*

We thank the reviewer for their evaluation of the manuscript, and for appreciating our caution in the identification of SIP occurrences. We also believe that one contribution of this work is to investigate the possible occurrence of SIP beyond the columnar growth region.

We fully agree with the reviewer that this case study provides material for the evaluation of model parameterization. In fact, we did conduct some preliminary experiments to assess the microphysical parameterization of WRF by comparing some model outputs to aircraft in-situ measurements. However, this brings several new challenges (as seen for example in Young et al., 2019). One challenge is that mismatches between aircraft observations and modeled results may come not only from the microphysical description but also from e.g., imperfect wind or advection resulting in precipitation cells being slightly misplaced in spaced or time —which significantly affects the comparison with pointwise aircraft measurements. Another example of challenge is, that the aircraft measurements themselves are not straightforward to process for this comparison; for instance, the PSDs reconstructed from PIP and 2D-S measurements following Leroy et al. (2016) cannot separate ice and liquid particles. This severely hampers the computation of in-situ ICNC; if only larger particles are used to compute the ICNC, this naturally makes the comparison less relevant as the newly-formed ice crystals would primarily affect the smaller size bins. Addressing these challenges was beyond the scope of this case study, although further work may focus on this topic (based on this event or others from ICE GENESIS). We believe that another promising approach would involve the use of a higher resolution model with bin microphysics or emulated bin frameworks (Zhao et al., 2021) e.g., to capture more finely the diversity of ice species reflected by the Doppler multimodalities, as well as the quite localized aspect of certain features that we observed (e.g., during phase III). We included a sentence in the conclusion of the revised manuscript to highlight possible future research directions:

As highlighted by e.g., Sinclair et al. (2016); Young et al. (2019), such case studies where SIP is presumed to be active may also serve to evaluate and improve the microphysical parameterization of SIP processes within numerical weather models. Along this line, further work may include more modeling-oriented approaches, (...).

1. *P6 line 170. Your notation for the primary ice mode as rimer, implies that riming is occurring in all the cases. Is that correct and applicable for the whole event? If not, than another name would be better, for example Verlinde et al. (2013) refer to such particles as background ice. Not sure if this is the best name, but at least it does not imply any processes.*

   We thank the reviewer for pointing this out and for the suggestion. While riming is indeed occurring during most of the event, this is not the primary focus of the study and there may be some time steps where it is is not dominant. Since we would also like to keep the idea that these particles are precipitating from above, and that they are possibly riming when LW is present, while sticking to a simple objective description, we adopted the wording of Oue et al. (2015, 2018) and now refer to this mode as "faster-falling" particles. This was changed throughout the text and in the figures.

   (...) the *primary mode*, sometimes referred to as the *rimer* (Kalesse et al., 2016) or *faster-falling* mode (Oue et al., 2015, 2018, which is the wording used hereafter), denotes the peak with the largest Doppler velocity
* * *
2. *P 12. Figure 4. The bright-band that can be observed starting from 1530 UTC just above 500 m, is it the melting layer? If it is, the panel a). needs to be adjusted. Could you please clarify this.*

   Thank you for raising this point. This is an interesting feature related to the onset of the warm front: the bright band corresponds to a partial melting layer, below which the particles freeze again (confirmed by aircraft in situ temperature measurements). Therefore, the classification should remain valid also below this layer; within the partial melting layer itself, a high SLDR may be observed that corresponds to melting particles; but below the layer, a high SLDR is the sign of prolate particles. Nonetheless, as the altitude range below this bright band is not the focus of this study, we decided to gray it out in panel a) of Fig. 4 to avoid any risk of confusion.

   See new Fig. 4.
* * *
3. *p.20. line 453 and below, So WRF was also used to understand the origin of SLW. You may want to point it out when you describe the WRF simulations.*

   Indeed, this aspect was missing in the explanation of how WRF simulations are used in the study. The paragraph now reads:

   The WRF simulations are used in this study to provide high-resolution temperature, wind and humidity profiles, to gain an understanding of the mesoscale processes and how they may contribute to snowfall microphysics over LCDF. The model is also used to investigate the mechanisms that sustain mixed-phase conditions during the event (see Sect. 5.2.3 (new Sect. 5.3.3)).
* * *
4. *p.21 line 491-493 Changes in MDV of the rimer mode is a combination of changes in air motion and particle properties, as manifested by observed fall-streaks in the reflectivity field (Fig. 3). Because of this MDV of the rimer mode may not be the best suited for estimating EDR. Did you consider using velocity of the liquid peak? That is why in Li et al. (2021) the liquid peak velocity was used to study air motion.*

We agree that using the velocity of the liquid peak in the computation of the EDR based on Shupe et al. (2008) would be the ideal solution. Unfortunately, the liquid peak was only rarely distinctly visible in the spectra (at the altitude we are focusing on during this phase), most likely because of the turbulent broadening itself. This means that it was not feasible to compute the variance of its velocity (required to derive the EDR), as this would require to have, at several range gates, valid velocity values for a large number of timesteps. We added a sentence to clarify this choice:

Note that $\text{MDV}_{F3}$ is used here for lack of more robust information: ideally, the EDR should be computed from the variance of the MDV of the liquid mode, but the latter is only rarely distinctly visible and thus cannot be used.
* * *
5. *In the conclusions you are stating that more involved multi-sensor approaches should be used to confirm occurrence of SIP, could you explain what you mean. Your study mainly relies on Doppler radar observations. The aircraft data was just used to confirm that your inference is not wrong. So what sensors do you miss and need to make your analysis more conclusive?*

We prefer to be cautious in the interpretations we propose of SIP processes: the radar measurements cannot directly prove that a given mechanism is taking place; rather, we can say that they concur with the hypotheses we formulate. The fact that the aircraft observations lower down are also compatible strongly supports our inferences. Nonetheless, we can in principle not exclude that slightly different processes—perhaps not fully known yet—would yield similar signatures. This sentence of the conclusion is intended to highlight that for an actual proof that a process is occurring, direct in-situ measurements at the location of the new ice production may be relevant; also, INP and ICNC measurements exactly at this location and at the appropriate temperature would help confirm the hypotheses. We tried to clarify:

All in all, the interpretation of these processes remains hypothetical; the fact that both the radar signatures and the aircraft observations lower down are compatible with the proposed explanations strongly supports these inferences. However, an unambiguous demonstration of the occurrence of SIP via a specific process is a challenge that would require more in-situ measurements across scales, and in the precise temperature range where the crystals are being formed, to get a full picture of ICNC, of INP availability, and of the interactions between ice (and liquid) particles.

**References**

Aydin, K. and Tang, C. (1997). Relationships between IWC and polarimetric radar measurands at 94 and 220 GHz for hexagonal columns and plates. *Journal of Atmospheric and Oceanic Technology*, 14(5):1055–1063.

Billault-Roux, A.-C., Grazioli, J., Delanoë, J., Jorquera, S., Pauwels, N., Viltard, N., Martini, A., Mariage, V., Le Gac, C., Caudoux, C., Aubry, C., Bertrand, F., Schwarzenboeck, A., Jaffeux, L., Coutris, P., Febvre, G., Pichon, J. M., Dezitter, F., Gehring, J., Untersee, A., Calas, C., Figueras i Ventura, J., Vie, B., Peyrat, A., Curat, V., Rebouissoux, S., and Berne, A. (2023). ICE GENE-SIS: Synergetic Aircraft and Ground-Based Remote Sensing and In Situ Measurements of Snowfall Microphysical Properties. *Bulletin of the American Meteorological Society*, 104(2):E367–E388.

Frisch, A. S., Fairall, C. W., and Snider, J. B. (1995). Measurement of Stratus Cloud and Drizzle Parameters in ASTEX with a Ka-Band Doppler Radar and a Microwave Radiometer. *Journal of the Atmospheric Sciences*, 52(16):2788–2799.

Garrett, T. J., Fallgatter, C., Shkurko, K., and Howlett, D. (2012). Fall speed measurement and high-resolution multi-angle photography of hydrometeors in free fall. *Atmospheric Measurement Techniques*, 5(11):2625–2633.

Hartmann, S., Seidel, J., Keinert, A., Kiselev, A., Leisner, T., and Stratmann, F. (2023). Secondary ice production - No evidence of a productive rime-splintering mechanism during dry and wet growth. In *EGU General Assembly*, Vienna.

Heymsfield, A. J. and Baumgardner, D. (1985). Summary of a Workshop on Processing 2-D Probe Data. *Bulletin of the American Meteorological Society*, 66(4):437–440.

Jaffeux, L., Schwarzenböck, A., Coutris, P., and Duroure, C. (2022). Ice crystal images from optical array probes: Classification with convolutional neural networks. *Atmospheric Measurement Techniques*, 15(17):5141–5157.

Joe, P. and List, R. (1987). Testing and Performance of Two-Dimensional Optical Array Spectrometers with Greyscale. *Journal of Atmospheric and Oceanic Technology*, 4(1):139–150.

Kalesse, H., Szyrmer, W., Kneifel, S., Kollias, P., and Luke, E. (2016). Fingerprints of a riming event on cloud radar Doppler spectra: Observations and modeling. *Atmospheric Chemistry and Physics*, 16(5):2997–3012.

Kogan, Z. N., Mechem, D. B., and Kogan, Y. L. (2005). Assessment of variability in continental low stratiform clouds based on observations of radar reflectivity. *Journal of Geophysical Research D: Atmospheres*, 110(18):1–15.

Korolev, A. (2007). Reconstruction of the sizes of spherical particles from their shadow images. Part I: Theoretical considerations. *Journal of Atmospheric and Oceanic Technology*, 24(3):376–389.

Korolev, A. and Leisner, T. (2020). Review of experimental studies of secondary ice production. *Atmospheric Chemistry and Physics*, 20(20):11767–11797.

Leinonen, J., Kneifel, S., Moisseev, D., Tyynelä, J., Tanelli, S., and Nousiainen, T. (2012). Evidence of nonspheroidal behavior in millimeter-wavelength radar observations of snowfall. *Journal of Geophysical Research Atmospheres*, 117(17):1–10.

Leroy, D., Fontaine, E., Schwarzenboeck, A., and Strapp, J. W. (2016). Ice Crystal Sizes in High Ice Water Content Clouds. Part I: On the Computation of Median Mass Diameter from In Situ Measurements. *Journal of Atmospheric and Oceanic Technology*, 33(11):2461–2476.

Li, H., Korolev, A., and Moisseev, D. (2021). Supercooled liquid water and secondary ice production in Kelvin-Helmholtz instability as revealed by radar Doppler spectra observations. *Atmospheric Chemistry and Physics*, 21(17):13593–13608.

Li, H. and Moisseev, D. (2019). Melting Layer Attenuation at Ka- and W-Bands as Derived From Multifrequency Radar Doppler Spectra Observations. *Journal of Geophysical Research: Atmospheres*, 124(16):9520–9533.

Li, H. and Moisseev, D. (2020). Two Layers of Melting Ice Particles Within a Single Radar Bright Band: Interpretation and Implications. *Geophysical Research Letters*, 47(13).

Luke, E. P., Yang, F., Kollias, P., Vogelmann, A. M., and Maahn, M. (2021). New insights into ice multiplication using remote-sensing observations of slightly supercooled mixed-phase clouds in the Arctic. *Proceedings of the National Academy of Sciences of the United States of America*, 118(13):1–9.

Matrosov, S. Y. (1991). Theoretical Study of Radar Polarization Parameters Obtained from Cirrus Clouds. *Journal of the Atmospheric Sciences*, 48(8):1062–1070.

Matrosov, S. Y., Mace, G. G., Marchand, R., Shupe, M. D., Hallar, A. G., and Mccubbin, I. B. (2012). Observations of ice crystal habits with a scanning polarimetric W-band radar at slant linear depolarization ratio mode. *Journal of Atmospheric and Oceanic Technology*, 29(8):989–1008.

Matrosov, S. Y., Reinking, R. F., Kropfli, R. A., Martner, B. E., and Bartram, B. W. (2001). On the Use of Radar Depolarization Ratios for Estimating Shapes of Ice Hydrometeors in Winter Clouds. *Journal of Applied Meteorology*, 40(3):479–490.

Mech, M., Maahn, M., Kneifel, S., Ori, D., Orlandi, E., Kollias, P., Schemann, V., and Crewell, S. (2020). PAMTRA 1.0: the Passive and Active Microwave radiative TRAnsfer tool for simulating radiometer and radar measurements of the cloudy atmosphere. *Geoscientific Model Development*, 13(9):4229–4251.

Morrison, H., van Lier-Walqui, M., Fridlind, A. M., Grabowski, W. W., Harrington, J. Y., Hoose, C., Korolev, A., Kumjian, M. R., Milbrandt, J. A., Pawlowska, H., Posselt, D. J., Prat, O. P., Reimel, K. J., Shima, S., van Diedenhoven, B., and Xue, L. (2020). Confronting the Challenge of Modeling Cloud and Precipitation Microphysics. *Journal of Advances in Modeling Earth Systems*, 12(8):e2019MS001689.

Myagkov, A., Seifert, P., Bauer-Pfundstein, M., and Wandinger, U. (2016). Cloud radar with hybrid mode towards estimation of shape and orientation of ice crystals. *Atmospheric Measurement Techniques*, 9(2):469–489.

Ori, D., von Terzi, L., Karrer, M., and Kneifel, S. (2021). snowScatt 1.0: consistent model of microphysical and scattering properties of rimed and unrimed snowflakes based on the self-similar Rayleigh–Gans approximation. *Geoscientific Model Development*, 14(3):1511–1531.

Oue, M., Kollias, P., Ryzhkov, A., and Luke, E. P. (2018). Toward Exploring the Synergy Between Cloud Radar Polarimetry and Doppler Spectral Analysis in Deep Cold Precipitating Systems in the Arctic. *Journal of Geophysical Research: Atmospheres*, 123(5):2797–2815.

Oue, M., Kumjian, M. R., Lu, Y., Verlinde, J., Aydin, K., and Clothiaux, E. E. (2015). Linear depolarization ratios of columnar ice crystals in a deep precipitating system over the arctic observed by zenith-pointing Ka-band doppler radar. *Journal of Applied Meteorology and Climatology*, 54(5):1060–1068.

Oue, M., Tatarevic, A., Kollias, P., Wang, D., Yu, K., and Vogelmann, A. M. (2020). The Cloud-resolving model Radar SIMulator (CR-SIM) Version 3.3: description and applications of a virtual observatory. *Geoscientific Model Development*, 13(4):1975–1998.

Pfitzenmaier, L., Dufournet, Y., Unal, C. M., and Russchenberg, H. W. (2017). Retrieving fall streaks within cloud systems using doppler radar. *Journal of Atmospheric and Oceanic Technology*, 34(4):905–920.

Pfitzenmaier, L., Unal, C. M., Dufournet, Y., and Russchenberg, H. W. (2018). Observing ice particle growth along fall streaks in mixed-phase clouds using spectral polarimetric radar data. *Atmospheric Chemistry and Physics*, 18(11):7843–7862.

Rambukkange, M. P., Verlinde, J., Eloranta, E. W., Flynn, C. J., and Clothiaux, E. E. (2011). Using Doppler Spectra to Separate Hydrometeor Populations and Analyze Ice Precipitation in Multilayered Mixed-Phase Clouds. *IEEE Geoscience and Remote Sensing Letters*, 8(1):108–112.

Ramelli, F., Henneberger, J., David, R. O., Buehl, J., Radenz, M., Seifert, P., Wieder, J., Lauber, A., Pasquier, J. T., Engelmann, R., Mignani, C., Hervo, M., and Lohmann, U. (2021). Microphysical investigation of the seeder and feeder region of an Alpine mixed-phase cloud. *Atmospheric Chemistry and Physics*, 21(9):6681–6706.

Reinking, R. F., Matrosov, S. Y., Kropfli, R. A., and Bartram, B. W. (2002). Evaluation of a 45° Slant Quasi-Linear Radar Polarization State for Distinguishing Drizzle Droplets, Pristine Ice Crystals, and Less Regular Ice Particles. *Journal of Atmospheric and Oceanic Technology*, 19(3):296–321.

Ryzhkov, A. V. and Zrnic, D. S. (2019). *Radar polarimetry for weather observations*. Springer Atmospheric Sciences.

Shupe, M. D., Kollias, P., Matrosov, S. Y., and Schneider, T. L. (2004). Deriving Mixed-Phase Cloud Properties from Doppler Radar Spectra. *Journal of Atmospheric and Oceanic Technology*, 21(4):660–670.

Shupe, M. D., Kollias, P., Poellot, M., and Eloranta, E. (2008). On deriving vertical air motions from cloud radar doppler spectra. *Journal of Atmospheric and Oceanic Technology*, 25(4):547–557.

Sinclair, V. A., Moisseev, D., and Von Lerber, A. (2016). How dual-polarization radar observations can be used to verify model representation of secondary ice. *Journal of Geophysical Research*, 121(18):10,954–10,970.

Tridon, F., Battaglia, A., and Kneifel, S. (2020). Estimating total attenuation using Rayleigh targets at cloud top: Applications in multilayer and mixed-phase clouds observed by ground-based multifrequency radars. *Atmospheric Measurement Techniques*, 13(9):5065–5085.

Vaillant De Guélis, T., Schwarzenböck, A., Shcherbakov, V., Gourbeyre, C., Laurent, B., Dupuy, R., Coutris, P., and Duroure, C. (2019). Study of the diffraction pattern of cloud particles and the respective responses of optical array probes. *Atmospheric Measurement Techniques*, 12(4):2513–2529.

Verlinde, J., Rambukkange, M. P., Clothiaux, E. E., McFarquhar, G. M., and Eloranta, E. W. (2013). Arctic multilayered, mixed-phase cloud processes revealed in millimeter-wave cloud radar Doppler spectra. *Journal of Geophysical Research Atmospheres*, 118(23):13,199–13,213.

von Terzi, L., Dias Neto, J., Ori, D., Myagkov, A., and Kneifel, S. (2022). Ice microphysical processes in the dendritic growth layer: a statistical analysis combining multi-frequency and polarimetric Doppler cloud radar observations. *Atmospheric Chemistry and Physics*, 22(17):11795–11821.

Young, G., Lachlan-Cope, T., O'Shea, S. J., Dearden, C., Listowski, C., Bower, K. N., Choularton, T. W., and Gallagher, M. W. (2019). Radiative Effects of Secondary Ice Enhancement in Coastal Antarctic Clouds. *Geophysical Research Letters*, 46(4):2312–2321.

Zawadzki, I., Fabry, F., and Szyrmer, W. (2001). Observations of supercooled water and secondary ice generation by a vertically pointing X-band Doppler radar. *Atmospheric Research*, 59-60:343–359.

Zhao, X., Liu, X., Phillips, V. T. J., and Patade, S. (2021). Impacts of secondary ice production on arctic mixed-phase clouds based on arm observations and cam6 single-column model simulations. *Atmospheric Chemistry and Physics*, 21(7):5685–5703.